# Improved Gradient based Adversarial Attacks for Quantized Networks

## Abstract

Neural network quantization has become increasingly popular due to efficient memory consumption and faster computation resulting from bitwise operations on the quantized networks. Even though they exhibit excellent generalization capabilities, their robustness properties are not well-understood. In this work, we systematically study the robustness of quantized networks against gradient based adversarial attacks and demonstrate that these quantized models suffer from gradient vanishing issues and show a fake sense of robustness. By attributing gradient vanishing to poor forward-backward signal propagation in the trained network, we introduce a simple temperature scaling approach to mitigate this issue while preserving the decision boundary. Despite being a simple modification to existing gradient based adversarial attacks, experiments on CIFAR-10/100 datasets with multiple network architectures demonstrate that our temperature scaled attacks obtain near-perfect success rate on quantized networks while outperforming original attacks on adversarially trained models as well as floating-point networks.

## 1 Introduction

Neural Network (NN) quantization has become increasingly popular due to reduced memory and time complexity enabling real-time applications and inference on resource-limited devices. Such quantized networks often exhibit excellent generalization capabilities despite having low capacity due to reduced precision for parameters and activations. However, their robustness properties are not well-understood. In particular, while parameter quantized networks are claimed to have better robustness against gradient based adversarial attacks (Galloway et al. (2018)), activation only quantized methods are shown to be vulnerable (Lin et al. (2019)).

In this work, we consider the extreme case of Binary Neural Networks (BNNs) and systematically study the robustness properties of parameter quantized models, as well as both parameter and activation quantized models against gradient based adversarial attacks. Our analysis reveals that these quantized models suffer from gradient masking issues (Athalye et al. (2018)) (especially vanishing gradients) and in turn show fake robustness. We attribute this vanishing gradients issue to poor forward-backward signal propagation caused by trained binary weights, and our idea is to improve signal propagation of the network without affecting the prediction of the classifier.

There is a body of work on improving signal propagation in a neural network (*e.g.*, Glorot & Bengio (2010); Pennington et al. (2017); Lu et al. (2020)), however, we are facing a unique challenge of *improving signal propagation while preserving the decision boundary*, since our ultimate objective is to generate adversarial attacks. To this end, we first discuss the conditions to ensure informative gradients and then resort to a temperature scaling approach (Guo et al. (2017)) (which scales the logits before applying softmax cross-entropy) to show that, even with a single positive scalar the vanishing gradients issue in BNNs can be alleviated achieving *near perfect success rate* in all tested cases.

Specifically, we introduce two techniques to choose the temperature scale: 1) based on the singular values of the input-output Jacobian, 2) by maximizing the norm of the Hessian of the loss with respect to the input. The justification for the first case is that if the singular values of input-output Jacobian are concentrated around 1 (defined as dynamical isometry (Pennington et al. (2017))) then the network is said to have good signal propagation and we intend to make the mean of singular

values to be 1. On the other hand, the intuition for maximizing the Hessian norm is that if the Hessian norm is large, then the gradient of the loss with respect to the input is sensitive to an infinitesimal change in the input. This is a sufficient condition for the network to have good signal propagation as well as informative gradients under the assumption that the network does not have any randomized or non-differentiable components.

We evaluated our improved gradient based adversarial attacks using BNNs with weight quantized (BNN-WQ) and weight and activation quantized (BNN-WAQ), floating point networks (REF), and adversarially trained models. We employ quantized and floating point networks trained on CIFAR-10/100 datasets using several architectures. In all tested BNNs, both versions of our temperature scaled attacks obtained near-perfect success rate outperforming gradient based attacks (FGSM (Goodfellow et al. (2014)), PGD (Madry et al. (2017))). Furthermore, this temperature scaling improved gradient based attacks even on adversarially trained models (both high-precision and quantized) as well as floating point networks, showing the significance of signal propagation for adversarial attacks.

## 2    PRELIMINARIES

We first provide some background on the neural network quantization and adversarial attacks.

### 2.1    NEURAL NETWORK QUANTIZATION

Neural Network (NN) quantization is defined as training networks with parameters constrained to a minimal, discrete set of quantization levels. This primarily relies on the hypothesis that since NNs are usually overparametrized, it is possible to obtain a quantized network with performance comparable to the floating point network. Given a dataset $\mathcal{D} = \{\mathbf{x}_i, \mathbf{y}_i\}_{i=1}^n$, NN quantization can be written as:

$$\min_{\mathbf{w} \in \mathcal{Q}^m} L(\mathbf{w}; \mathcal{D}) := \frac{1}{n} \sum_{i=1}^n \ell(\mathbf{w}; (\mathbf{x}_i, \mathbf{y}_i)) \, . \tag{1}$$

Here, $\ell(\cdot)$ denotes the input-output mapping composed with a standard loss function (*e.g.*, cross-entropy loss), $\mathbf{w}$ is the $m$ dimensional parameter vector, and $\mathcal{Q}$ is a predefined discrete set representing quantization levels (*e.g.*, $\mathcal{Q} = \{-1, 1\}$ in the binary case).

Most of the NN quantization approaches (Ajanthan et al. (2019a;b); Bai et al. (2019); Hubara et al. (2017)) convert the above problem into an unconstrained problem by introducing auxiliary variables and optimize via (stochastic) gradient descent. To this end, the algorithms differ in the choice of quantization set (*e.g.*, keep it discrete (Courbariaux et al. (2015)), relax it to the convex hull (Bai et al. (2019)) or convert the problem into a lifted probability space (Ajanthan et al. (2019a))), the projection used, and how differentiation through projection is performed. In the case when the constraint set is relaxed, a gradually increasing annealing hyperparameter is used to enforce a quantized solution (Ajanthan et al. (2019a;b); Bai et al. (2019)). We refer the interested reader to respective papers for more detail. In this paper, we use BNN-WQ obtained using MD-tanh-S (Ajanthan et al. (2019b)) and BNN-WAQ obtained using Hubara et al. (2017).

### 2.2    ADVERSARIAL ATTACKS

Adversarial examples consist of imperceptible perturbations to the data that alter the model's prediction with high confidence. Existing attacks can be categorized into white-box and black-box attacks where the difference lies in the knowledge of the adversaries. White-box attacks allow the adversaries access to the target model's architecture and parameters, whereas black-box attacks can only query the model. Since white-box gradient based attacks are popular, we summarize them below.

First-order gradient based attacks can be compactly written as Projected Gradient Descent (PGD) on the negative of the loss function (Madry et al. (2017)). Formally, let $\mathbf{x}^0 \in \mathbb{R}^N$ be the input image, then at iteration $t$, the PGD update can be written as:

$$\mathbf{x}^{t+1} = P\left(\mathbf{x}^t + \eta\, \mathbf{g}_\mathbf{x}^t\right) \, , \tag{2}$$

where $P : \mathbb{R}^N \to \mathcal{X}$ is a projection, $\mathcal{X} \subset \mathbb{R}^N$ is the constraint set that bounds the perturbations, $\eta > 0$ is the step size, and $\mathbf{g}_\mathbf{x}^t$ is a form of gradient of the loss with respect to the input $\mathbf{x}$ evaluated at $\mathbf{x}^t$. With this general form, the popular gradient based adversarial attacks can be specified:

- **Fast Gradient Sign Method (FGSM)**: This is a one step attack introduced in Goodfellow et al. (2014). Here, $P$ is the identity mapping, $\eta$ is the maximum allowed perturbation magnitude, and

| Method | ResNet-18 | | | VGG-16 | | |
|---|---|---|---|---|---|---|
| | **Clean** | **Adv.(1)** | **Adv.(20)** | **Clean** | **Adv.(1)** | **Adv.(20)** |
| **REF** | 94.46 | 0.00 | 0.00 | 93.31 | 0.04 | 0.00 |
| **BNN-WQ** | 93.18 | 26.98 | 17.91 | 91.53 | 47.32 | 38.49 |
| **BNN-WAQ** | 87.67 | 8.57 | 1.94 | 89.69 | 78.01 | 59.26 |

Table 1: *Clean and adversarial accuracy (PGD attack with $L_\infty$ bound) on the test set of CIFAR-10 using ResNet-18 and VGG-16. In brackets, we mention number of random restarts used to perform the attack. Note,* BNN*s outperform adversarial accuracy of floating point networks consistently.*

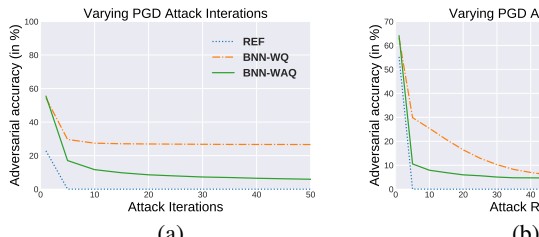
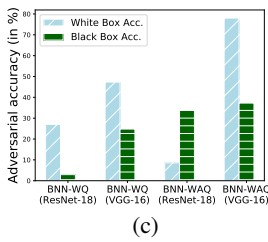

|          (a)          |          (b)          |          (c)          |

Figure 1: *Gradient marking checks in ResNet-18 on CIFAR-10 for* PGD *attack with $L_\infty$ bound: (a) varying iterations, (b) varying radius, and (c) black-box attacks on ResNet-18 and VGG-16. While (a), (c) show signs of gradient masking, (b) does not. We attribute this discrepancy to the random initial step before* PGD.

$\mathbf{g_x}^t = \text{sign}\left(\nabla_\mathbf{x}\ell(\mathbf{w}^*;(\mathbf{x}^t,\mathbf{y}))\right)$, where $\ell$ denotes the loss function, $\mathbf{w}^*$ is the trained weights and $\mathbf{y}$ is the ground truth label corresponding to the image $\mathbf{x}^0$.

- **PGD with $L_\infty$ bound**: Arguably the most popular adversarial attack introduced in Madry et al. (2017) and sometimes referred to as Iterative Fast Gradient Sign Method (IFGSM). Here, $P$ is the $L_\infty$ norm based projection, $\eta$ is a chosen step size, and $\mathbf{g_x}^t = \text{sign}\left(\nabla_\mathbf{x}\ell(\mathbf{w}^*;(\mathbf{x}^t,\mathbf{y}))\right)$, the sign of gradient same as FGSM.
- **PGD with $L_2$ bound**: This is also introduced in Madry et al. (2017) which performs the standard PGD in the Euclidean space. Here, $P$ is the $L_2$ norm based projection, $\eta$ is a chosen step size, and $\mathbf{g_x}^t = \nabla_\mathbf{x}\ell(\mathbf{w}^*;(\mathbf{x}^t,\mathbf{y}))$ is simply the gradient of the loss with respect to the input.

These attacks have been further strengthened by a random initial step (Tramèr et al. (2017)). In this paper, we perform this single random initialization for all experiments with FGSM/PGD attack unless otherwise mentioned.

## 3 ROBUSTNESS EVALUATION OF BINARY NEURAL NETWORKS

We start by evaluating the adversarial accuracy (i.e. accuracy on the perturbed data) of BNNs using the PGD attack with $L_\infty$ bound.

- **PGD attack details**: perturbation bound of 8 pixels (assuming each pixel in the image is in $[0, 255]$) with respect to $L_\infty$ norm, step size $\eta = 2$ and the total number of iterations $T = 20$. The attack details are the same in all evaluated settings unless stated otherwise.

We perform experiments on CIFAR-10 dataset using ResNet-18 and VGG-16 architectures and report the clean accuracy and PGD adversarial accuracy with 1 and 20 random restarts in Table 1. It can be clearly and consistently observed that binary networks have high adversarial accuracy compared to the floating point counterparts. Even with 20 random restarts, BNNs clearly outperform floating point networks in terms of adversarial accuracy. Since this result is surprising, we investigate this phenomenon further to understand whether BNNs are actually robust to adversarial perturbations or they show a fake sense of security due to some form of obfuscated gradients (Athalye et al. (2018)).

### 3.1 IDENTIFYING OBFUSCATED GRADIENTS

Recently, it has been shown that several defense mechanisms intentionally or unintentionally break gradient descent and cause obfuscated gradients and thus exhibit a false sense of security (Athalye et al. (2018)). Several gradient based adversarial attacks tend to fail to produce adversarial perturbations in scenarios where the gradients are uninformative, referred to as gradient masking. Gradient masking

can occur due to shattered gradients, stochastic gradients or exploding and vanishing gradients. We try to identify gradient masking in binary networks based on the empirical checks provided in Athalye et al. (2018). If any of these checks fail, it indicates gradient masking issue in BNNs.

To illustrate this, we analyse the effects of varying different hyperparameters of PGD attack on BNNs trained on CIFAR-10 using ResNet-18 architecture. Even though varying PGD perturbation bound does not show any signs of gradient masking, varying attack iterations and black-box vs white-box results (on ResNet-18 and VGG-16) clearly indicate gradient masking issues as depicted in Fig. 1. The black-box attack outperforming white-box attack for BNNs certainly indicates gradient masking issues since the black-box attack do not use the gradient information from model being attacked. Here, our black-box model to a BNN is the analogous floating point network trained on the same dataset and the attack is the same PGD with $L_\infty$ bound.

These checks demonstrate that BNNs are prone to gradient masking and exhibit fake robustness. Note, shattered gradients occur due to non-differentiable components in the defense mechanism and stochastic gradients are caused by randomized gradients. Since BNNs are trainable from scratch and does not have randomized gradients[1], we narrow down gradient masking issue to vanishing or exploding gradients. Since, vanishing or exploding gradients occur due to poor signal propagation, by introducing a single scalar, we discuss two approaches to mitigate this issue, which lead to almost 100% success rate for gradient based attacks on BNNs.

## 4 SIGNAL PROPAGATION OF NEURAL NETWORKS

We first describe how poor signal propagation in neural networks can cause vanishing or exploding gradients. Then we discuss the idea of introducing a single scalar to improve the existing gradient based attacks without affecting the prediction (*i.e.*, decision boundary) of the trained models.

We consider a neural network $f_\mathbf{w}$ for an input $\mathbf{x}^0$, having logits $\mathbf{a}^K = f_\mathbf{w}(\mathbf{x}^0)$. Now, since softmax cross-entropy is usually used as the loss function, we can write:

$$\ell(\mathbf{a}^K, \mathbf{y}) = -\mathbf{y}^T \log(\mathbf{p}), \qquad \mathbf{p} = \text{softmax}(\mathbf{a}^K), \tag{3}$$

where $\mathbf{y} \in \mathbb{R}^d$ is the one-hot encoded target label and $\log$ is applied elementwise.

For various gradient based adversarial attacks discussed in Sec. 2.2, gradient of the loss $\ell$ is used with respect to the input $\mathbf{x}^0$, which can also be formulated using chain rule as,

$$\frac{\partial \ell(\mathbf{a}^K, \mathbf{y})}{\partial \mathbf{x}^0} = \frac{\partial \ell(\mathbf{a}^K, \mathbf{y})}{\partial \mathbf{a}^K} \frac{\partial \mathbf{a}^K}{\partial \mathbf{x}^0} = \psi(\mathbf{a}^K, \mathbf{y}) \mathbf{J}, \tag{4}$$

where $\psi$ denotes the error signal and $\mathbf{J} \in \mathbb{R}^{d \times N}$ is the input-output Jacobian. Here we use the convention that $\partial \mathbf{v} / \partial \mathbf{u}$ is of the form $\mathbf{v}$-size $\times$ $\mathbf{u}$-size.

Notice there are two components that influence the gradients, 1) the Jacobian $\mathbf{J}$ and 2) the error signal $\psi$. Gradient based attacks would fail if either the Jacobian is poorly conditioned or the error signal has saturating gradients, both of these will lead to vanishing gradients in $\partial \ell / \partial \mathbf{x}^0$.

The effects of Jacobian on the signal propagation is studied in dynamical isometry and mean-field theory literature (Pennington et al. (2017); Saxe et al. (2013)) and it is known that a network is said to satisfy dynamical isometry if the singular values of $\mathbf{J}$ are concentrated near 1. Under this condition, error signals $\psi$ backpropagate isometrically through the network, approximately preserving its norm and all angles between error vectors. Thus, as dynamical isometry improves the trainability of the floating point networks, a similar technique can be useful for gradient based attacks as well.

In fact, almost all initialization techniques (*e.g.*, Glorot & Bengio (2010)) approximately ensures that the Jacobian $\mathbf{J}$ is well-conditioned for better trainability and it is hypothesized that approximate isometry is preserved even at the end of the training. But, for BNNs, the weights are constrained to be $\{-1, 1\}$ and hence the weight distribution at end of training is completely different from the random initialization. Furthermore, it is not clear that fully-quantized networks can achieve well-conditioned Jacobian, which guided some research activity in utilizing layerwise scalars (either predefined or learned) to improve BNN training (McDonnell (2018); Rastegari et al. (2016)). We would like to point out that the focus of this paper is to improve gradient based attacks on already trained BNNs. To

---

[1] BNN-WQ have binary weights, but there is no non-differentiable or randomized component once trained.

this end learning a new scalar to improve signal propagation at each layer is not useful as it can alter the decision boundary of the network and thus cannot be used in practice on already trained model.

### 4.1 TEMPERATURE SCALING FOR BETTER SIGNAL PROPAGATION

In this paper, we propose to use a single scalar per network to improve the signal propagation of the network using temperature scaling. In fact, one could replace softmax with a monotonic function such that the prediction is not altered, however, we will show in our experiments that a single scalar with softmax has enough flexibility to improve signal propagation and yields almost $100\%$ success rate with PGD attacks. Essentially, we can use a scalar, $\beta > 0$ without changing the decision boundary of the network by preserving the relative order of the logits. Precisely, we consider the following:

$$\mathbf{p}(\beta) = \text{softmax}(\bar{\mathbf{a}}^K) , \qquad \bar{\mathbf{a}}^K = \beta \, \mathbf{a}^K . \tag{5}$$

Here, we write the softmax output probabilities $\mathbf{p}$ as a function of $\beta$ to emphasize that they are softmax output of temperature scaled logits. Now since in this context, the only variable is the temperature scale $\beta$, we denote the loss and the error signal as functions of only $\beta$. With this simplified notation the gradient of the temperature scaled loss with respect to the inputs can be written as:

$$\frac{\partial \ell(\beta)}{\partial \mathbf{x}^0} = \frac{\partial \ell(\beta)}{\partial \bar{\mathbf{a}}^K} \frac{\partial \bar{\mathbf{a}}^K}{\partial \mathbf{a}^K} \frac{\partial \mathbf{a}^K}{\partial \mathbf{x}^0} = \psi(\beta)\beta \, \mathbf{J} . \tag{6}$$

Note that $\beta$ affects the input-output Jacobian linearly while it nonlinearly affects the error signal $\psi$. To this end, we hope to obtain a $\beta$ that ensures the error signal is useful (*i.e.*, not all zero) as well as the Jacobian is well-conditioned to allow the error signal to propagate to the input.

We acknowledge that while one can find a $\beta > 0$ to obtain softmax output ranging from a uniform distribution ($\beta = 0$) to one-hot vectors ($\beta \to \infty$), $\beta$ only scales the Jacobian. Therefore, if the Jacobian $\mathbf{J}$ has zero singular values, our approach has no effect in those dimensions. However, since most of the modern networks consist of ReLU nonlinearities (generally positive homogeneous functions), the effect of a single scalar would be equivalent (ignoring the biases) to having layerwise scalars such as in McDonnell (2018). Thus, we believe a single scalar is sufficient for our purpose.

## 5 IMPROVED GRADIENTS FOR ADVERSARIAL ATTACKS

Now we discuss strategies to choose a scalar $\beta$ such that the gradients with respect to input are informative. Let us first analyze the effect of $\beta$ on the error signal. To this end,

$$\psi(\beta) = \frac{\partial \ell(\beta)}{\partial \mathbf{p}(\beta)} \frac{\partial \mathbf{p}(\beta)}{\partial \bar{\mathbf{a}}^K} = -(\mathbf{y} - \mathbf{p}(\beta))^T . \tag{7}$$

where $\mathbf{y}$ is the one-hot encoded target label, and $\mathbf{p}(\beta)$ is the softmax output of scaled logits.

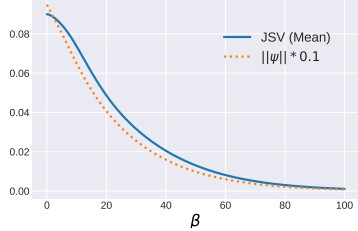

Figure 2: *Error signal ($\psi(\beta)$) and Jacobian of softmax ($\partial \mathbf{p}(\beta)/\partial \bar{\mathbf{a}}^K$) vs. $\beta$ for a random correctly classified logits.*

For adversarial attacks, we only consider the correctly classified images (*i.e.*, $\arg\max_j y_j = \arg\max_j p_j(\beta)$) as there is no need to generate adversarial examples corresponding to misclassified samples. From the above formula, it is clear that when $\mathbf{p}(\beta)$ is one-hot encoding then the error signal is $\mathbf{0}$. This is one of the reason for vanishing gradient issue in BNNs. Even if this does not happen for a given image, one can increase $\beta \to \infty$ to make this error signal $\mathbf{0}$. Similarly, when $\mathbf{p}(\beta)$ is the uniform distribution, the norm of the error signal is at the maximum. This can be obtained by setting $\beta = 0$. However, this would also make $\partial \ell(\beta)/\partial \mathbf{x}^0 = \mathbf{0}$ as the singular values of the input-output Jacobian would all be 0. How error signal is affected by $\beta$ is illustrated in Fig. 2.

This analysis indicates that the optimal $\beta$ cannot be obtained by simply maximizing the norm of the error signal and we need to balance both the Jacobian as well as the error signal. To summarize, the scalar $\beta$ should be chosen such that the following properties are satisfied:

1. $\|\psi(\beta)\|_2 > \rho$ for some $\rho > 0$.
2. The Jacobian $\beta \mathbf{J}$ is well-conditioned, *i.e.*, the singular values of $\beta \mathbf{J}$ is concentrated around 1.

### 5.1 NETWORK JACOBIAN SCALING (NJS)

We now discuss a straightforward, two-step approach to attain the aforementioned properties. Firstly, to ensure $\beta \mathbf{J}$ is well-conditioned, we simply choose $\beta$ to be the inverse of the mean of singular values

of $\mathbf{J}$. This guarantees that the mean of singular values of $\beta\mathbf{J}$ is 1. After this scaling, it is possible that the resulting error signal is very small. To ensure that $\|\psi(\beta)\|_2 > \rho > 0$, we ensure that the softmax output $p_k(\beta)$ corresponding to the ground truth class $k$ is at least $\rho$ away from 1. We now state it as a proposition to derive $\beta$ given a lowerbound on $1 - p_k(\beta)$.

**Proposition 1.** Let $\mathbf{a}^K \in \mathbb{R}^d$ with $d > 1$ and $a_1^K \geq a_2^K \geq \ldots \geq a_d^K$ and $a_1^K - a_d^K = \gamma$. For a given $0 < \rho < (d-1)/d$, there exists a $\beta > 0$ such that $1 - \text{softmax}(\beta a_1^K) > \rho$, then $\beta < -\log(\rho/(d-1)(1-\rho))/\gamma$.

*Proof.* This is derived via a simple algebraic manipulation of softmax. Please refer to Appendix. $\square$

This $\beta$ can be used together with the one computed using inverse of mean Jacobian Singular Values (JSV). We provide the pseudocode for our proposed PGD++ (NJS) attack in Appendix. Similar approach can also be applied for FGSM++. Notice that, this approach is simple and it adds negligible overhead to the standard PGD attacks. However, it has a hyperparameter $\rho$ which is hand designed. To mitigate this, next we discuss a hyperparameter-free approach to obtain $\beta$.

## 5.2 HESSIAN NORM SCALING (HNS)

We now discuss another approach to obtain informative gradients. Our idea is to maximize the Frobenius norm of the Hessian of the loss with respect to the input, where the intuition is that if the Hessian norm is large, then the gradient $\partial\ell/\partial\mathbf{x}^0$ is sensitive to an infinitesimal change in $\mathbf{x}^0$. This means, the infinitesimal perturbation in the input is propagated in the forward pass to the last layer and propagated back to the input layer without attenuation (*i.e.*, the returned signal is not zero), assuming there are no randomized or non-differentiable components in the network. This clearly indicates that the network has good signal propagation as well as the error signals are not all zero. This objective can now be written as:

$$\beta^* = \underset{\beta>0}{\text{argmax}} \left\| \frac{\partial^2\ell(\beta)}{\partial(\mathbf{x}^0)^2} \right\|_F = \underset{\beta>0}{\text{argmax}} \left\| \beta\left[\psi(\beta)\frac{\partial\mathbf{J}}{\partial\mathbf{x}^0} + \beta\left(\frac{\partial\mathbf{p}(\beta)}{\partial\bar{\mathbf{a}}^K}\mathbf{J}\right)^T\mathbf{J}\right] \right\|_F. \tag{8}$$

The derivation is provided in Appendix. Note, since $\mathbf{J}$ does not depend on $\beta$, $\mathbf{J}$ and $\partial\mathbf{J}/\partial\mathbf{x}^0$ are computed only once, $\beta$ is optimized using grid search as it involves only a single scalar. In fact, it is easy to see from the above equation that, when the Hessian is maximized, $\beta$ cannot be zero. Similarly, $\psi(\beta)$ cannot be zero because if it is zero, then the prediction $\mathbf{p}(\beta)$ is one-hot encoding (Eq. (7)), consequently $\partial\mathbf{p}(\beta)/\partial\bar{\mathbf{a}}^K = \mathbf{0}$ and this cannot be a maximum for the Hessian norm. Hence, this ensures that $\|\psi(\beta^*)\|_2 > \rho$ for some $\rho > 0$ and $\beta^*$ is bounded according to Proposition 1. Therefore, the maximum is obtained for a finite value of $\beta$. Even though, it is not clear how exactly this approach

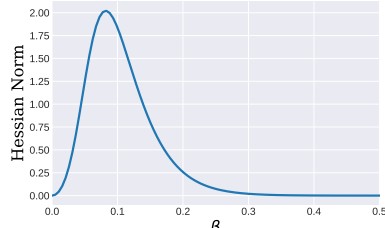

Figure 3: *Hessian norm vs. $\beta$ on a random correctly classified image. The plot clearly shows a concave behaviour. s*

would affect the singular values of the input-output Jacobian ($\beta\mathbf{J}$), we know that they are finite and not zero. How Hessian norm is influenced by $\beta$ is illustrated in Fig. 3.

Furthermore, there are some recent works (Moosavi-Dezfooli et al. (2019); Qin et al. (2019)) show that adversarial training makes the loss surface locally linear around the vicinity of training samples and enforcing local linearity constraint on loss curvature can achieve better robust to adversarial attacks. On the contrary, our idea of maximizing the Hessian, *i.e.*, increasing the nonlinearity of $\ell$, could make the network more prone to adversarial attacks and we intend to exploit that. The psuedocode for PGD++ attack with HNS is summarized in Appendix.

## 6 EXPERIMENTS

We evaluate robustness accuracies of BNNs with weight quantized (BNN-WQ), weight and activation quantized (BNN-WAQ) floating point networks (REF), and adversarially trained networks. We evaluate our two PGD++ variants corresponding to Hessian Norm Scaling (HNS) and Network Jacobian Scaling (NJS) on CIFAR-10 and CIFAR-100 datasets with multiple network architectures. Briefly, our results indicate that both of our proposed attack variants yield attack success rate much higher than original PGD attacks not only on $L_\infty$ bounded attack but also on $L_2$ bounded attacks on both floating point networks and binarized networks. Our proposed PGD++ variants also reduce PGD

| | Network | Adversarial Accuracy (%) | | | | | | | | |
|---|---|---|---|---|---|---|---|---|---|---|
| | | FGSM | FGSM++ | | PGD ($L_\infty$) | PGD++ ($L_\infty$) | | PGD ($L_2$) | PGD++ ($L_2$) | |
| | | | NJS | HNS | | NJS | HNS | | NJS | HNS |
| CIFAR-10 | ResNet-18 | 40.49 | 3.46 | **2.51** | 26.98 | **0.00** | **0.00** | 74.59 | **0.05** | **0.05** |
| | VGG-16 | 57.55 | 4.00 | **3.43** | 47.32 | **0.00** | **0.00** | 61.90 | **0.35** | 1.32 |
| | ResNet-50 | 57.62 | 6.44 | **5.35** | 43.14 | **0.00** | **0.00** | 74.75 | 0.11 | **0.08** |
| | DenseNet-121 | 26.80 | 4.67 | **4.24** | 9.11 | **0.00** | **0.00** | 72.99 | **0.03** | 0.06 |
| | MobileNet-V2 | 33.50 | 6.42 | **5.42** | 26.86 | **0.00** | **0.00** | 35.22 | 0.12 | **0.09** |
| CIFAR-100 | ResNet-18 | 25.22 | 14.08 | **1.80** | 8.23 | 2.45 | **0.00** | 42.67 | 6.79 | **0.26** |
| | VGG-16 | 19.82 | 7.98 | **1.76** | 17.44 | 0.88 | **0.16** | 19.26 | 3.17 | **0.63** |
| | ResNet-50 | 37.76 | 16.33 | **14.17** | 25.71 | **2.33** | 2.73 | 38.95 | 7.9 | **7.41** |
| | DenseNet-121 | 28.32 | 12.21 | **10.86** | 8.87 | 1.15 | **1.09** | 43.78 | 4.54 | **4.16** |
| | MobileNet-V2 | 12.09 | 10.18 | **8.79** | 1.44 | **0.57** | 0.66 | 8.97 | 3.39 | **3.01** |

Table 2: *Adversarial accuracy on the test set for* BNN-WQ. *Both our* NJS *and* HNS *variants consistently outperform original* $L_\infty$ *bounded* FGSM *and* PGD *attack, and* $L_2$ *bounded* PGD *attack.*

| | Network | Adversarial Accuracy (%) | | | | | | | | |
|---|---|---|---|---|---|---|---|---|---|---|
| | | FGSM | FGSM++ | | PGD ($L_\infty$) | PGD++ ($L_\infty$) | | PGD ($L_2$) | PGD++ ($L_2$) | |
| | | | NJS | HNS | | NJS | HNS | | NJS | HNS |
| REF | ResNet-18 | 7.62 | 5.55 | **5.35** | **0.00** | **0.00** | **0.00** | 45.18 | 0.09 | **0.05** |
| | VGG-16 | 11.01 | 10.04 | **9.66** | 0.04 | **0.00** | **0.00** | 2.23 | **0.78** | 1.10 |
| | ResNet-50 | 21.64 | 6.08 | **5.70** | 0.69 | **0.00** | **0.00** | 65.56 | **0.07** | 0.09 |
| | DenseNet-121 | 11.40 | 7.58 | **7.30** | **0.00** | **0.00** | **0.00** | 38.15 | 0.08 | **0.06** |
| BNN-WAQ | ResNet-18 | 40.84 | 19.46 | **19.09** | 8.57 | **0.03** | 0.04 | 67.84 | **2.33** | 2.59 |
| | VGG-16 | 79.92 | 15.96 | **15.39** | 78.01 | **0.01** | 0.02 | 85.62 | **0.49** | 0.62 |
| | ResNet-50 | 33.16 | **25.89** | 27.05 | 0.49 | **0.23** | 0.45 | 32.93 | **6.68** | 8.77 |
| | DenseNet-121 | 37.20 | **23.89** | 24.69 | 0.81 | **0.10** | 0.18 | 59.32 | **3.72** | 6.17 |

Table 3: *Adversarial accuracy on the test set of CIFAR-10 for* REF *and* BNN-WAQ. *Both our* NJS *and* HNS *variants consistently outperform original* FGSM *and* PGD *($L_\infty/L_2$ bounded) attacks.*

adversarial accuracy of adversarially trained floating point and adversarially trained binarized neural networks while outperforming much stronger attacks such as DeepFool (Moosavi-Dezfooli et al. (2016)) and Brendel & Bethge Attack (BBA) (Brendel et al. (2019)). Among our variants, even though they perform similarly in our experiments, Hessian based scaling (HNS) outperforms Jacobian based scaling (NJS) in majority of the cases and this difference is significant for one step FGSM attacks. This indicates that nonlinearity of the network indeed has some relationship to its adversarial robustness.

We use state of the art models trained for binary quantization (where all layers are quantized) for our experimental evaluations. We provide adversarial attack parameters used for FGSM/PGD in Appendix and for other attacks, we use default parameters used in Foolbox (Rauber et al. (2017)). For our HNS variant, we sweep $\beta$ from a range such that the hessian norm is maximized for each image, as explained in Appendix. For our NJS variant, we set the value of $\rho = 0.01$. In fact, our attacks are not very sensitive to $\rho$ and we provide the ablation study in the Appendix. The PyTorch (Paszke et al. (2017)) implementation of our algorithm will be released upon publication.

## 6.1 RESULTS

We first compared the original PGD ($L_2/L_\infty$) and FGSM attack with both versions (NJS and HNS) of improved PGD++ and FGSM++ attack, on CIFAR-10/100 datasets with ResNet-18/50, VGG-16, DenseNet-121 and MobileNet-V2 network architectures and the adversarial accuracies for different BNN-WQ are reported in Table 2. Our PGD++ variants consistently outperform original PGD on all networks on both datasets. Even being a gradient based attack, our proposed PGD++ ($L_2/L_\infty$) variants can in fact reach adversarial accuracy close to 0 on CIFAR-10 dataset, demystifying the fake robustness binarized networks tend to exhibit due to poor signal propagation.

| Network | Adversarial Accuracy (%) | | | | | | | | | |
|---------|------|----------------------|------|------|------|-------------------|-----------|------|------|------|
| | FGSM | FGSM $\beta = 0.1$ | FGSM++ | | PGD | PGD $\beta = 0.1$ | Deep Fool | BBA | PGD++ | |
| | | | NJS | HNS | | | | | NJS | HNS |
| REF | 62.38 | 69.52 | 61.43 | **61.40** | 48.73 | 61.27 | 51.01 | 48.43 | **47.17** | 48.54 |
| BC | 53.91 | 62.46 | 52.90 | **52.27** | 41.29 | 54.24 | 42.65 | 40.14 | 39.35 | **39.34** |
| GD-tanh | 56.13 | 65.06 | 55.54 | **54.81** | 42.77 | 56.78 | 44.78 | 42.94 | **42.14** | 42.30 |
| MD-tanh-S | 55.10 | 63.42 | 54.74 | **53.82** | 41.34 | 54.22 | 43.46 | 40.69 | 40.76 | **40.67** |

Table 4: *Adversarial accuracy on the test set of CIFAR-10 with ResNet-18 for adversarially trained* REF *and* BNN-WQ *using different quantization methods (*BC, GD-tanh, MD-tanh-S*). Our improved attacks are compared against* FGSM, $L_\infty$ *bounded* PGD, *a heuristic choice of* $\beta = 0.1$, *DeepFool and* BBA. *Albeit on adversarially trained networks, our methods outperform all the comparable methods.*

| Network | Adversarial Accuracy (%) | | | | | | | |
|---------|------|------------|--------|------|------|-----------|-------|------|
| | FGSM | FGSM (DLR) | FGSM++ | | PGD | PGD (DLR) | PGD++ | |
| | | | NJS | HNS | | | NJS | HNS |
| REF | 7.62 | 19.48 | 5.55 | **5.35** | 0.00 | 0.00 | 0.00 | 0.00 |
| BNN-WQ | 40.49 | 19.72 | 3.46 | **2.51** | 26.98 | 0.00 | 0.00 | 0.00 |
| BNN-WAQ | 40.84 | 41.78 | 19.46 | **19.09** | 8.57 | 4.57 | **0.03** | 0.04 |
| REF[*] | 62.38 | 66.39 | 61.43 | **61.40** | 48.73 | 49.73 | **47.17** | 48.54 |
| BNN-WQ[*] | 55.10 | 59.14 | 54.74 | **53.82** | 41.34 | 41.42 | 40.76 | **40.67** |

Table 5: *Adversarial accuracy for* REF, BNN-WQ, *and* BNN-WAQ *trained on CIFAR-10 using ResNet-18. Here* [*] *denotes adversarially trained models. Both our* NJS *and* HNS *variants consistently outperform* $L_\infty$ *bounded* FGSM *and* PGD *attack performed with Difference of Logits Ratio (*DLR*) loss instead of cross entropy loss. Notice,* FGSM *and* PGD *attack with* DLR *loss (Croce & Hein (2020)) perform even worse than their original form on adversarially trained models.*

Similarly, for one step FGSM attack, our modified versions outperform original FGSM attacks by a significant margin consistently for both datasets on various network architectures. We would like to point out such an improvement in the above two attacks is considerably interesting, knowing the fact that FGSM, PGD with $L_\infty$ attacks only use the sign of the gradients so improved performance indicates, our temperature scaling indeed makes some zero elements in the gradient nonzero. We would like to point out here that one can use several random restarts to increase the success rate of original form of FGSM/PGD attack further but to keep comparisons fair we use single random restart for both original and modified attacks. Nevertheless, as it has been observed in Table 1 even with 20 random restarts PGD adversarial accuracies for BNNs cannot reach zero, whereas our proposed PGD++ variants consistently achieve perfect success rate.

**ImageNet.** For other large scale datasets such as ImageNet, BNNs are hard to train with full binarization of parameters and result in poor performance. Thus, most existing works (Yang et al. (2019)) on BNNs keep the first and the last layers floating point and introduce several layerwise scalars to achieve good results on ImageNet. In such experimental setups, according to our experiments, trained BNNs do not exhibit gradient masking issues or poor signal propagation and thus are easier to attack using original FGSM/PGD attacks with complete success rate. In such experiments, our modified versions perform equally well compared to the original forms of these attacks.

The adversarial accuracies of REF and BNN-WAQ trained on CIFAR-10 using ResNet-18/50, VGG-16 and DenseNet-121 for our variants against original counterparts are reported in Table 3. Overall, for both REF and BNN-WAQ, our variants outperform the original counterparts consistently. Particularly interesting, PGD++ variants improve the attack success rate on REF networks. This effectively expands the applicability of our PGD++ variants and encourages to consider signal propagation of any trained network to improve gradient based attacks. PGD++ with $L_\infty$ variants achieve near-perfect success rate on all BNN-WAQs, again validating the hypotheses of fake robustness of BNNs.

To further demonstrate the efficacy, we first adversarially trained the BNN-WQs (quantized using BC (Courbariaux et al. (2015)), GD-tanh/MD-tanh-S (Ajanthan et al. (2019b))) and floating point networks in a similar manner as in Madry et al. (2017), using $L_\infty$ bounded PGD with $T = 7$

iterations, $\eta = 2$ and $\epsilon = 8$. We report the adversarial accuracies of $L_\infty$ bounded attacks and our variants on CIFAR-10 using ResNet-18 in Table 4. These results further strengthens the usefulness of our proposed PGD++ variants. Moreover, with a heuristic choice of $\beta = 0.1$ to scale down the logits before performing gradient based attacks performs even worse. Finally, even against stronger attacks (DeepFool (Moosavi-Dezfooli et al. (2016)), BBA (Brendel et al. (2019))) under the same $L_\infty$ perturbation bound, our variants outperform consistently on these adversarially trained models. We would like to point out that our variants have negligible computational overhead over the original gradient based attacks, whereas stronger attacks are much slower in practice requiring 100-1000 iterations with an adversarial starting point (instead of random initial perturbation).

To illustrate the effectiveness of our proposed variants in improving signal propagation, we compare against gradient based attacks performed using recently proposed Difference of Logits Ratio (DLR) loss (Croce & Hein (2020)) that aims to avoid the issue of saturating error signals. We show these experimental comparisons performed on ResNet-18 models trained on CIFAR-10 dataset in Table 5. The attack parameters are same as used for the other experiments. It can be clearly observed that in almost all cases our proposed variants are much better than original form of gradient based attacks performed with DLR loss. The margin of difference is significant in case of FGSM attack and adversarial trained models. Infact, it is important to note that gradient based attacks with DLR loss perform worse on adversarially trained models than the original form of gradient based attacks.

## 7 RELATED WORK

Adversarial examples are first observed in Szegedy et al. (2014) and subsequently efficient gradient based attacks such as FGSM (Goodfellow et al. (2014)) and PGD (Madry et al. (2017)) are introduced. There exist recent stronger attacks such as Moosavi-Dezfooli et al. (2016); Carlini & Wagner (2017); Yao et al. (2019); Finlay et al. (2019); Brendel et al. (2019), however, compared to PGD, they are much slower to be used for adversarial training in practice. For a comprehensive survey related to adversarial attacks, we refer the reader to Chakraborty et al. (2018).

Some recent works focus on the adversarial robustness of BNNs (Bernhard et al. (2019); Sen et al. (2020); Galloway et al. (2018); Khalil et al. (2019); Lin et al. (2019)), however, a strong consensus on the robustness properties of quantized networks is lacking. In particular, while Galloway et al. (2018) claims parameter quantized networks are robust to gradient based attacks based on empirical evidence, (Lin et al. (2019)) shows activation quantized networks are vulnerable to such attacks and proposes a defense strategy assuming the parameters are floating-point. Differently, Khalil et al. (2019) proposes a combinatorial attack hinting that activation quantized networks would have obfuscated gradients issue. Sen et al. (2020) shows ensemble of mixed precision networks to be more robust than original floating point networks; however Tramer et al. (2020) later shows the presented defense method can be attacked with minor modification in the loss function. In short, although it has been hinted that there might be some sort of gradient masking in BNNs (especially in activation quantized networks), a thorough understanding is lacking on whether BNNs are robust, if not what is the reason for the inferior performance of most commonly used gradient based attacks on binary networks. We answer this question in this paper and introduce improved gradient based attacks.

## 8 CONCLUSION

In this work, we have shown that both BNN-WQ and BNN-WAQ tend to show a fake sense of robustness on gradient based attacks due to poor signal propagation. To tackle this issue, we introduced our two variants of PGD++ attack, namely NJS and HNS. Our proposed PGD++ variants not only possess near-complete success rate on binarized networks but also outperform standard $L_\infty$ and $L_2$ bounded PGD attacks on floating point networks. We finally show improvement in attack success rate on adversarially trained REF and BNN-WQ against stronger attacks (DeepFool and BBA). In future, we intend to focus more on improving the robustness of the BNNs with provable robustness guarantees.

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

# Appendices

Here, we first provide the pseudocodes, proof of the proposition and the derivation of Hessian. Later we give additional experiments, analysis and the details of our experimental setting.

## A    PSEUDOCODE

We provide pseudocode for PGD++ with NJS in Algorithm 1 and PGD++ with HNS in Algorithm 2.

---

**Algorithm 1** PGD++ with NJS with $L_\infty$, $T$ iterations, radius $\epsilon$, step size $\eta$, network $f_{\mathbf{w}^*}$, input $\mathbf{x}^0$, label $k$, one-hot $\mathbf{y} \in \{0,1\}^d$, gradient threshold $\rho$.

---

**Require:** $T, \epsilon, \eta, \rho, \mathbf{x}^0, \mathbf{y}, k$
**Ensure:** $\|\mathbf{x}^{T+1} - \mathbf{x}^0\|_\infty \leq \epsilon$
1:   $\beta_1 = (M\,d)/\left(\sum_{i=1}^M \sum_{j=1}^d \mu_j(\mathbf{J}_i)\right)$               $\triangleright$ $\beta_1$ computed using Network Jacobian.
2:   $\mathbf{x}^1 = P_\infty^\epsilon(\mathbf{x}^0 + \text{Uniform}(-1,1))$               $\triangleright$ Random Initialization with Projection
3:   **for** $t \leftarrow 1, \dots T$ **do**
4:       $\beta_2 = 1.0$
5:       $\mathbf{p}' = \text{softmax}(\beta_1(f_{\mathbf{w}^*}(\mathbf{x}^t)))$
6:       **if** $1 - p_k' \leq \rho$ **then**               $\triangleright$ $\rho = 0.01$
7:           $\beta_2 = -\log(\rho/(d-1)(1-\rho))/\gamma$               $\triangleright$ $\gamma$ computed using Proposition 2
8:       $\ell = -\mathbf{y}^T \log(\text{softmax}(\beta_2\beta_1(f_{\mathbf{w}^*}(\mathbf{x}^t))))$
9:       $\mathbf{x}^{t+1} = P_\infty^\epsilon(\mathbf{x}^t + \eta\,\text{sign}(\nabla_{\mathbf{x}}\ell(\mathbf{x}^t)))$               $\triangleright$ Update Step with Projection

---

**Algorithm 2** PGD++ with HNS with $L_\infty$, $T$ iterations, radius $\epsilon$, step size $\eta$, network $f_{\mathbf{w}^*}$, input $\mathbf{x}^0$, label $k$, one-hot $\mathbf{y} \in \{0,1\}^d$, gradient threshold $\rho$.

---

**Require:** $T, \epsilon, \eta, \mathbf{x}^0, \mathbf{y}, k$
**Ensure:** $\|\mathbf{x}^{T+1} - \mathbf{x}^0\|_\infty \leq \epsilon$
1:   $\mathbf{x}^1 = P_\infty^\epsilon(\mathbf{x}^0 + \text{Uniform}(-1,1))$               $\triangleright$ Random Initialization with Projection
2:   $\beta^* = \text{argmax}_{\beta>0} \left\|\partial^2\ell(\beta)/\partial(\mathbf{x}^0)^2\right\|_F$               $\triangleright$ Grid Search
3:   **for** $t \leftarrow 1, \dots T$ **do**
4:       $\ell = -\mathbf{y}^T \log(\text{softmax}(\beta^*(f_{\mathbf{w}^*}(\mathbf{x}^t))))$
5:       $\mathbf{x}^{t+1} = P_\infty^\epsilon(\mathbf{x}^t + \eta\,\text{sign}(\nabla_{\mathbf{x}}\ell(\mathbf{x}^t)))$               $\triangleright$ Update Step with Projection

---

## B    DERIVATIONS

### B.1    DERIVING $\beta$ GIVEN A LOWERBOUND ON $1 - p_k(\beta)$

**Proposition 2.** Let $\mathbf{a}^K \in \mathbb{R}^d$ with $d > 1$ and $a_1^K \geq a_2^K \geq \ldots \geq a_d^K$ and $a_1^K - a_d^K = \gamma$. For a given $0 < \rho < (d-1)/d$, there exists a $\beta > 0$ such that $1 - \text{softmax}(\beta a_1^K) > \rho$, then $\beta < -\log(\rho/(d-1)(1-\rho))/\gamma$.

*Proof.* Assuming $a_1^K - a_d^K = \gamma$, we derive a condition on $\beta$ such that $1 - \mathrm{softmax}(\beta a_1^K) > \rho$.

$$1 - \mathrm{softmax}(\beta a_1^K) > \rho \,, \tag{9}$$

$$\mathrm{softmax}(\beta a_1^K) < 1 - \rho \,,$$

$$\exp(\beta a_1^K) / \sum_{\lambda=1}^{d} \exp(\beta a_\lambda^K) < 1 - \rho \,,$$

$$1 / \big(1 + \sum_{\lambda=2}^{d} \exp(\beta(a_\lambda^K - a_1^K))\big) < 1 - \rho \,.$$

Since, $a_1^K - a_\lambda^K \leq \gamma$ for all $\lambda > 1$,

$$1 / \big(1 + \sum_{\lambda=2}^{d} \exp(\beta(a_\lambda^K - a_1^K))\big) \leq 1 / \big(1 + \sum_{\lambda=2}^{d} \exp(-\beta\gamma)\big) \,. \tag{10}$$

Therefore, to ensure $1 / \big(1 + \sum_{\lambda=2}^{d} \exp(\beta(a_\lambda^K - a_1^K))\big) < 1 - \rho$, we consider,

$$1 / \big(1 + \sum_{\lambda=2}^{d} \exp(-\beta\gamma)\big) < 1 - \rho \,, \quad a_1^K - a_\lambda^K \leq \gamma \text{ for all } \lambda > 1 \,, \tag{11}$$

$$1 / \big(1 + (d-1)\exp(-\beta\gamma)\big) < 1 - \rho \,,$$

$$\exp(-\beta\gamma) > \rho/(d-1)(1-\rho) \,,$$

$$-\beta\gamma > \log(\rho/(d-1)(1-\rho)) \,, \quad \exp \text{ is monotone} \,,$$

$$\beta < -\log(\rho/(d-1)(1-\rho))/\gamma \,.$$

Therefore for any $\beta < -\log(\rho/(d-1)(1-\rho))/\gamma$, the above inequality $1 - \mathrm{softmax}(\beta a_1^K) > \rho$ is satisfied. $\qquad\square$

## B.2 DERIVATION OF HESSIAN

We now derive the Hessian of the input mentioned in Eq. (8) of the paper. The input gradients can be written as:

$$\frac{\partial \ell(\beta)}{\partial \mathbf{x}^0} = \frac{\partial \ell(\beta)}{\partial \mathbf{p}(\beta)} \frac{\partial \mathbf{p}(\beta)}{\partial \bar{\mathbf{a}}^K(\beta)} \beta \mathbf{J} = \psi(\beta)\beta \mathbf{J} \,. \tag{12}$$

Now by product rule of differentiation, input hessian can be written as:

$$\frac{\partial^2 \ell(\beta)}{\partial (\mathbf{x}^0)^2} = \beta \left[ \psi(\beta)\frac{\partial \mathbf{J}}{\partial \mathbf{x}^0} + \left(\frac{\partial \psi(\beta)}{\partial \mathbf{x}^0}\right)^T \mathbf{J} \right] \,, \tag{13}$$

$$= \beta \left[ \psi(\beta)\frac{\partial \mathbf{J}}{\partial \mathbf{x}^0} + \left(\frac{\partial \mathbf{p}(\beta)}{\partial \mathbf{x}^0}\right)^T \mathbf{J} \right] \,, \quad \psi(\beta) = -(\mathbf{y} - \mathbf{p}(\beta))^T \,,$$

$$= \beta \left[ \psi(\beta)\frac{\partial \mathbf{J}}{\partial \mathbf{x}^0} + \beta \left(\frac{\partial \mathbf{p}(\beta)}{\partial \bar{\mathbf{a}}^K}\mathbf{J}\right)^T \mathbf{J} \right] \,.$$

## C ADDITIONAL EXPERIMENTS

In this section we first provide more experimental details and then some ablation studies.

## C.1 EXPERIMENTAL DETAILS

| Method | ResNet-18 | | | | VGG-16 | | | |
|---|---|---|---|---|---|---|---|---|
| | APGD | Square Attack | PGD++ | | APGD | Square Attack | PGD++ | |
| | | | NJS | HNS | | | NJS | HNS |
| REF | **0.00** | 0.55 | **0.00** | **0.00** | 0.79 | 2.25 | **0.00** | **0.00** |
| BNN-WQ | **0.00** | 0.41 | **0.00** | **0.00** | 8.23 | 1.98 | **0.00** | **0.00** |
| BNN-WAQ | 6.32 | 21.45 | **0.03** | 0.04 | 0.38 | 16.67 | **0.01** | 0.02 |

Table 7: *Adversarial accuracy for* REF, BNN-WQ *and* BNN-WAQ *trained on CIFAR-10 using ResNet-18. Both our* NJS *and* HNS *variants consistently outperform Auto-*PGD *(*APGD*) (Croce & Hein (2020)) performed using Difference of Logits Ratio (*DLR*) loss and a gradient free attack namely, Square Attack (Andriushchenko et al. (2020)) under $L_\infty$ bound (8/255).*

We first mention the hyperparameters used to perform FGSM and PGD attack for all the experiments in the paper in Table 6. To make a fair comparison, we keep the attack parameters same for our proposed variants of FGSM++ and PGD++ attacks. For PGD++ with HNS variant, we maximize Frobenius norm of Hessian with respect to the input as specified in Eq. (8) of the paper by grid search for the optimum $\beta$. We would like to point out that since only $\psi(\beta)$ and $\mathbf{p}(\beta)$ terms are dependent on $\beta$, we do not need to do forward and

| Dataset | Attack | $\epsilon$ | $\eta$ | $T$ |
|---|---|---|---|---|
| **CIFAR-10** | **FGSM** | 8 | 8 | 1 |
| | **PGD** $(L_\infty)$ | 8 | 2 | 20 |
| | **PGD** $(L_2)$ | 120 | 15 | 20 |
| **CIFAR-100** | **FGSM** | 4 | 4 | 1 |
| | **PGD** $(L_\infty)$ | 4 | 1 | 10 |
| | **PGD** $(L_2)$ | 60 | 15 | 10 |

Table 6: *Attack parameters ($\epsilon$ & $\eta$ in pixels).*

backward pass of the network multiple times during the grid search. This significantly reduces the computational overhead during the grid search. We can simply use the same network outputs $\mathbf{a}^K$ and network jacobian $\mathbf{J}$ (as computed without using $\beta$) for the grid search, while computing the other terms at each iteration of grid search. We apply grid search to find the optimum beta between 100 equally spaced intervals of $\beta$ starting from $\beta_1$ to $\beta_2$. Here, $\beta_1$ and $\beta_2$ are computed based on Proposition 1 in the paper where $\rho = 1e - 72$ and $\rho = 1 - (1/d) - (1e - 2)$ respectively, where $d$ is number of classes and $\gamma = a_1^K - a_2^K$ so that $1 - \text{softmax}(\beta a_1^K) < \rho$. Also, note that we estimate the optimum $\beta$ for each test sample only at the start of the first iteration of an iterative attack and then use the same $\beta$ for the next iterations.

**Computational Overhead of NJS and HNS.** Our Jacobian calculation takes just a single backward pass through the network and thus adds a negligible overhead. Our NJS approach for scaling estimates $\beta$ as inverse of mean JSV using 100 random test samples, which is similar to 100 backward passes. For HNS, in Eq. (8) Jacobian $\mathbf{J}$ can be computed in single backward pass. Moreover, for piecewise linear networks (eg, relu activations), $\partial \mathbf{J}/\partial \mathbf{x}^0 = 0$ almost everywhere (Yao et al. (2018)). Thus PGD++ with NJS and HNS is almost as efficient as PGD.

## C.2   COMPARISONS AGAINST AUTO-PGD ATTACK AND GRADIENT FREE ATTACK

We also compared our proposed PGD++ variants against recently proposed Auto-PGD (APGD) with Difference of Logits Ratio (DLR) loss (Croce & Hein (2020)) and gradient free Square Attack (Andriushchenko et al. (2020)) on different networks trained using ResNet-18 and VGG-16 on CIFAR-10 dataset and the results are reported in Table 7. The attack parameters for this experiment are the same as reported in the paper. It can be clearly seen that our proposed variants perform much better than both APGD with DLR loss and Square Attack, consistently achieving 0% adversarial accuracy. Infact, much computationally expensive Square attack is unable to achieve 0% adversarial accuracy in any of the cases under the enforced $L_\infty$ bound.

## C.3   OTHER EXPERIMENTS

We provide adversarial accuracy comparisons for different attack methods on CIFAR-100 using ResNet-18, VGG-16, ResNet-50 and DenseNet-121 in Table 8. Again similar to the results in the paper, our proposed PGD++ and FGSM++ outperform original form of PGD and FGSM consistently in all the experiments on floating point networks. We also provide adversarial accuracy comparison of our proposed variants against stronger attacks namely DeepFool (Moosavi-Dezfooli et al. (2016)) and

| Network | Adversarial Accuracy (%) | | | | | | | | |
|---------|------|------|------|-----------|----------|------|-----------|------|------|
| | FGSM | FGSM++ | | PGD ($L_\infty$) | PGD++ ($L_\infty$) | | PGD ($L_2$) | PGD++ ($L_2$) | |
| | | NJS | HNS | | NJS | HNS | | NJS | HNS |
| ResNet-18 | 9.06 | 9.23 | **2.70** | 0.14 | 0.14 | **0.00** | 5.38 | 0.17 | **0.15** |
| VGG-16 | 16.28 | 17.24 | **9.19** | 1.53 | 0.95 | **0.25** | 4.87 | 1.50 | **1.38** |
| ResNet-50 | 12.95 | 12.95 | **11.94** | 0.12 | **0.00** | **0.00** | 31.01 | 4.43 | **4.14** |
| DenseNet-121 | 11.41 | 11.41 | **10.74** | **0.00** | **0.00** | **0.00** | 6.10 | 3.09 | **2.76** |

Table 8: *Adversarial accuracy on the test set of CIFAR-100 for* REF *(floating point networks). Both our* NJS *and* HNS *variants consistently outperform original* FGSM *and* PGD *($L_\infty/L_2$ bounded) attacks.*

| Network | PGD | Deep Fool | BBA | PGD++ | |
|---------|-----|-----------|-----|-------|------|
| | | | | NJS | HNS |
| **ResNet-18** | 8.57 | 18.92 | 0.81 | **0.03** | 0.04 |
| **VGG-16** | 78.01 | 12.12 | 0.10 | **0.01** | 0.02 |

Table 9: *Adversarial accuracy on the test set of CIFAR-10 for* BNN-WAQ*. Here, we compare our proposed variants against much stronger attacks namely DeepFool (Moosavi-Dezfooli et al. (2016)) and* BBA *(Brendel et al. (2019)). Both our variants outperform stronger attacks. Note, DeepFool and* BBA *are much slower in practise requiring 100-1000 iterations.* BBA *specifically requires even an adversarial start point that needs to be computed using another adversarial attack.*

BBA (Brendel et al. (2019)) on BNN-WAQ trained on CIFAR-10 dataset in Table 9. In this experiment, our proposed variants again outperform even the stronger attacks which take 100-1000 iterations with adversarial start point (instead of random initial perturbation). It should be noted that although BBA performs much better than DeepFool and PGD, it still has inferior success rate than ours considering the fact that it takes multiple hours to run BBA whereas our proposed variants are almost as efficient as PGD attack.

**Step Size Tuning for PGD attack.** We would like to point out that step size $\eta$ and temperature scale $\beta$ have different effects in the attacks performed. Notice, PGD and FGSM attack under $L_\infty$ bound only use the sign of input gradients in each gradient ascent step. Thus, if the input gradients are completely saturated (which is the case for BNNs), original forms of PGD or FGSM will not work irrespective of the step size used. To illustrate this, we performed extensive step size tuning for original form of PGD attack on different ResNet-18 models trained on CIFAR-10 dataset and the adversarial accuracies are reported in Fig. 4. It can be observed clearly that although tuning the step size lowers adversarial accuracy a bit in some cases but still cannot reach zero for BNNs unlike our proposed variants.

**Adversarial training using PGD++.** We also investigate the potential application of PGD++ for adversarial training to improve the robustness of neural networks. PGD++ attack is most effective when applied to a network with poor signal propagation. However, adversarial training is performed from random initialization (Glorot & Bengio (2010)) exhibiting good signal propagation. Thus, PGD and PGD++ perform similarly for adversarial training. We infer these conclusions from our experiments on adversarial training using PGD++.

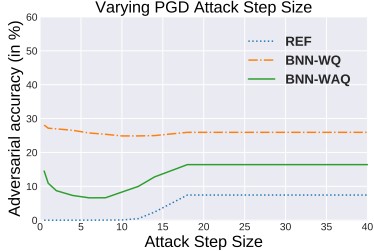

Figure 4: *Adversarial accuracy using* PGD *attack under $L_\infty$ bound (8/255) with varying step size ($\eta$) on ResNet-18 trained on CIFAR-10. Notice,* PGD *attack is unable to reach zero adversarial accuacy for* BNNs *with any step size.*

**CLEVER Scores.** Recently CLEVER Scores (Weng et al. (2018)) have been proposed as an empirical estimate to measure robustness lower bounds for deep networks. It has been later shown that gradient masking issues cause CLEVER to overestimate the robustness bounds (Goodfellow (2018)). Here we try to improve the CLEVER scores using different ways of choosing $\beta$ in temperature scaling. For this experiment, we use CLEVER implementation of Adversarial Training Toolbox[2] (Nicolae et al. (2018)). We set

---

[2]https://github.com/Trusted-AI/adversarial-robustness-toolbox

|  | Original | Heuristic | NJS | HNS |
|---|---|---|---|---|
| **BNN-WQ** | 0.8585 | 0.8845 | 0.4139 | **0.3450** |
| **BNN-WAQ** | 0.7239 | 3.1578 | 0.3120 | **0.2774** |

Table 10: CLEVER *Scores (Weng et al. (2018)) for* BNN-WQ *and* BNN-WAQ *trained on CIFAR-10 using ResNet-18. We compare CLEVER Scores returned for $L_1$ norm perturbation using different ways of temperature scaling applied. Here, Original refers to original network without temperature scaling and Heuristic denotes temperature scale with small $\beta = 0.01$.*

| **Methods** | **PGD++ (NJS) - Varying $\rho$** | | | | | |
|---|---|---|---|---|---|---|
|  | $1e-05$ | $1e-04$ | $1e-03$ | $1e-02$ | $1e-01$ | $2e-01$ |
| **REF** | 0.00 | 0.00 | 0.00 | 0.00 | 0.00 | 0.00 |
| **BNN-WQ** | 0.00 | 0.00 | 0.00 | 0.00 | 0.00 | 0.00 |
| **BNN-WAQ** | 0.15 | 0.08 | 0.04 | 0.03 | 0.04 | 0.02 |

Table 11: *Adversarial accuracy on the test set for binary neural networks using $L_\infty$ bounded PGD++ attack using* NJS *with varying $\rho$. For different values of $\rho$, our approach is quite stable.*

number of batches to 50, batch size to 10, radius to 5, and chose $L_1$ norm as hyperparameters (based on the Weng et al. (2018)). We compare our variants namely NJS and HNS against heuristic choice of small $\beta = 0.01$ and original CLEVER Scores for BNN-WQ and BNN-WAQ (trained on CIFAR-10 using ResNet-18) in Table 10. It can be clearly seen that our proposed variants improve the robustness bounds computed using CLEVER whereas a heuristic choice of $\beta = 0.01$ performs even worse.

## C.4    STABILITY OF PGD++ WITH NJS WITH VARIATIONS IN $\rho$

We perform ablation studies with varying $\rho$ for PGD++ with NJS in Table 11 for CIFAR-10 dataset using ResNet-18 architecture. It clearly illustrates that our NJS variant is quite robust to the choice of $\rho$ as we are able to achieve near perfect success rate with PGD++ with different values of $\rho$. As long as value of $\rho$ is large enough to avoid one-hot encoding on softmax outputs (in turn avoid $\|\psi(\beta)\|$ to be zero) of correctly classified sample, our approach with NJS variant is quite stable.

## C.5    SIGNAL PROPAGATION AND INPUT GRADIENT ANALYSIS USING NJS AND HNS

We first provide an example illustration in Fig. 5 to better understand how the input gradient norm *i.e.*, $\|\partial\ell(\beta)/\partial\mathbf{x}^0\|_2$, and norm of sign of input gradient, *i.e.*, $\|\text{sign}(\partial\ell(\beta)/\partial\mathbf{x}^0)\|_2$ is influenced by $\beta$. It clearly shows that both the plots have a concave behavior where an optimal $\beta$ can maximize the input gradient. Also, it can be quite evidently seen in Fig. 5 (b) that within an optimal range of $\beta$, gradient vanishing issue can be avoided. If $\beta \to 0$ or $\beta \to \infty$, it changes all the values in input gradient matrix to zero and inturn $\|\text{sign}(\partial\ell(\beta)/\partial\mathbf{x}^0)\|_2 = 0$.

We also provide the signal propagation properties as well as analysis on input gradient norm before and after using the $\beta$ estimated based on NJS and HNS in Table 12. For binarized networks as well floating point networks tested on CIFAR-10 dataset using ResNet-18 architecture, our HNS and NJS variants result in larger values for $\|\psi\|_2$, $\|\partial\ell(\beta)/\partial\mathbf{x}^0\|_2$ and $\|\text{sign}(\partial\ell(\beta)/\partial\mathbf{x}^0)\|_2$. This reflects the efficacy of our method in overcoming the gradient vanishing issue. It can be also noted that our variants also improves the signal propagation of the networks by bringing the mean JSV values closer to 1.

## C.6    ABLATION FOR $\rho$ VS. PGD++ ACCURACY

In this subsection, we provide the analysis on the effect of bounding the gradients of the network output of ground truth class $k$, *i.e.* $\partial\ell(\beta)/\partial\bar{a}_k^K$. Here, we compute $\beta$ using Proposition 1 for all correctly classified images such that $1 - \text{softmax}(\beta a_k^K) > \rho$ with different values of $\rho$ and report the PGD++ adversarial accuracy in Table 13. It can be observed that there is an optimum value of $\rho$ at which PGD++ success rate is maximized, especially on the adversarially trained models. This can also be seen in connection with the non-linearity of the network where at an optimum value of

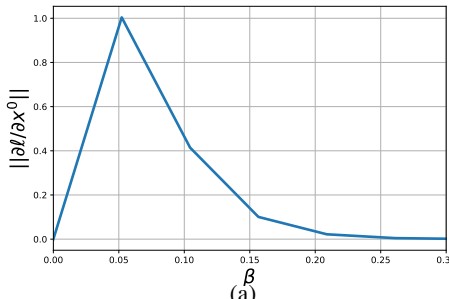
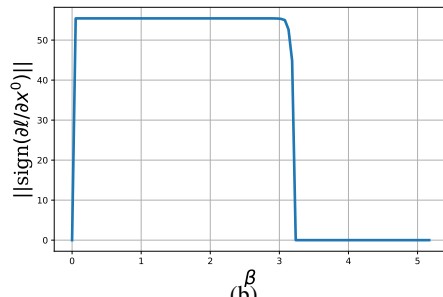

Figure 5: *Plots to show how variation in $\beta$ affects (a) norm of input gradient, i.e., $\|\partial\ell(\beta)/\partial\mathbf{x}^0\|_2$, (b) norm of sign of input gradient, i.e., $\|sign(\partial\ell(\beta)/\partial\mathbf{x}^0)\|_2$ on a random correctly classified image. Notice that, both input gradient and signed input gradient norm behave similarly, showing a concave behaviour. This plot is computed for* BNN-WQ *network on CIFAR-10, ResNet-18. (b) clearly illustrates how optimum $\beta$ can avoid vanishing gradient issue since $\|sign(\partial\ell(\beta)/\partial\mathbf{x}^0)\|_2$ will only be zero if input gradient matrix has only zeros.*

| Methods | | REF | Adv. Train | BNN-WQ | BNN-WAQ |
|---|---|---|---|---|---|
| **JSV (Mean)** | **Orig.** | 8.09e+00 | 5.15e−01 | 3.53e+01 | 1.11e+00 |
| | **NJS** | 9.51e−01 | 5.70e−01 | 9.95e−01 | 2.24e−01 |
| | **HNS** | 2.38e+00 | 6.11e+00 | 1.19e+01 | 4.65e+00 |
| **JSV (Std.)** | **Orig.** | 6.27e+00 | 4.10e−01 | 3.53e+01 | 1.97e+00 |
| | **NJS** | 7.58e−01 | 6.34e−01 | 9.71e−01 | 6.73e−01 |
| | **HNS** | 4.41e+00 | 5.34e+02 | 2.13e+02 | 1.24e+02 |
| $\|\psi\|_2$ | **Orig.** | 9.08e−03 | 2.33e−01 | 6.20e−03 | 9.46e−03 |
| | **NJS** | 4.66e−01 | 2.35e−01 | 5.37e−01 | 1.20e−01 |
| | **HNS** | 1.48e−01 | 2.57e−01 | 2.07e−01 | 2.44e−01 |
| $\|\partial\ell/\partial\mathbf{x}^0\|_2$ | **Orig.** | 2.42e−01 | 8.52e−02 | 2.27e−01 | 6.33e−02 |
| | **NJS** | 9.52e−01 | 1.10e−01 | 8.91e−01 | 1.24e−01 |
| | **HNS** | 7.49e−01 | 8.18e−01 | 3.70e−01 | 2.70e−01 |
| $\|sign\left(\frac{\partial\ell}{\partial\mathbf{x}^0}\right)\|_2$ | **Orig.** | 5.55e+01 | 5.54e+01 | 4.39e+01 | 5.55e+01 |
| | **NJS** | 5.55e+01 | 5.54e+01 | 5.55e+01 | 5.55e+01 |
| | **HNS** | 5.55e+01 | 5.54e+01 | 5.55e+01 | 5.55e+01 |

Table 12: *Mean and standard deviation of Jacobian Singular Values (*JSV*), mean $\|\psi\|_2$, mean $\|\partial\ell/\partial\mathbf{x}^0\|_2$ and mean $\|sign(\partial\ell/\partial\mathbf{x}^0)\|_2$ for different methods on CIFAR-10 with ResNet-18 computed with 500 correctly classified samples. Note here for* NJS *and* HNS*,* JSV *is computed for scaled jacobian i.e. $\beta\mathbf{J}$. Also note that, values of $\|\psi\|_2$, $\|\partial\ell(\beta)/\partial\mathbf{x}^0\|_2$ and $\|sign(\partial\ell(\beta)/\partial\mathbf{x}^0)\|_2$ are larger for our* NJS *and* HNS *variant (for most of the networks) as compared with network with no $\beta$, which clearly indicates better gradients for performing gradient based attacks.*

$\beta$, even for robust (locally linear) (Moosavi-Dezfooli et al. (2019); Qin et al. (2019)) networks such as adversarially trained models, non-linearity can be maximized and better success rate for gradient based attacks can be achieved. Our HNS variant essentially tries to achieve the same objective while trying to estimate $\beta$ for each example.

| Methods | PGD++ with Varying $\rho$ | | | | | |
|---|---|---|---|---|---|---|
| | $1e-15$ | $1e-09$ | $1e-05$ | $1e-01$ | $2e-01$ | $5e-01$ |
| **REF** | 0.00 | 0.00 | 0.00 | 0.00 | 0.00 | 0.00 |
| **BNN-WQ** | 9.61 | 0.04 | 0.00 | 0.00 | 0.00 | 0.00 |
| **REF**$^*$ | 48.18 | **47.66** | 48.00 | 53.09 | 54.58 | 57.57 |
| **BNN-WQ**$^*$ | 40.66 | **40.01** | 40.04 | 45.09 | 46.57 | 49.72 |

Table 13: *Adversarial accuracy on the test set for adversarially trained networks and binary neural networks using $L_\infty$ bounded* PGD++ *attack with varying $\rho$ as lower bound on the gradient of network output for ground truth class $k$. Here * denotes the adversarially trained models obtained where adversarial samples are generated using $L_\infty$ bounded* PGD *attack with with $T = 7$ iterations, $\eta = 2$ and $\epsilon = 8$. Note, here* PGD++ *attack refers to* PGD *attack where $\partial\ell(\beta)/\partial\bar{a}_k^K$ is bounded by $\rho$ for each sample, where $k$ is ground truth class.*

