# OpenReview forum: "Improved Gradient based Adversarial Attacks for Quantized Networks"
_ICLR.cc/2021/Conference — Reject_

### Official Review · AnonReviewer4 · 2020-10-28
**Some interesting techniques proposed, but contributions are too weak**

**Rating:** 6
**Confidence:** 5

**Review:**

**Update** : Since most of my issues have been addressed, I have changed my rating from 4 to 6

Summary:

This paper studies the robustness of quantized networks against gradient-based adversarial attacks (for L2 and Linf norms), showing how quantized models suffer from gradient vanishing, giving a false sense of security via gradient masking. To circumvent this issue, the authors propose temperature scaling approaches that can overcome this masking, achieving near-perfect perfect success in crafting adversarial inputs for these models.

##########################################################################

Reasons for score:

The paper's ultimate goal is to get better gradient-based attack performance on quantized (binarized, in this case) networks. However, key steps that should have been tried first for benchmarking such as adaptive PGD attacks have not been performed. Moreover, it is not clear what benefit the proposed method has in this scenario compared to gradient-free attacks like Boundary++.  The paper's contributions, although including some nice analyses on temperature scaling based solutions, are too weak to be accepted in their current form.


##########################################################################

Pros:

- Improvement in attack success rates for full-precision networks, even for FGSM, seems like an exciting result. Further analyses and methods on top of this could be used to further increase the strength of these first-order gradient attacks.

- Jacobian and Hessian based detailed analyses of temperature scaling, and what different solutions correspond to in terms of robustness is quite insightful and interesting.

##########################################################################

Cons:

- Gradient masking is a relatively well-known phenomenon in adversarial machine learning. In cases when normal first-order gradient attacks fail, techniques like adaptive PGD attacks, gradient-free attacks, or even black-box transfer attacks are some straightforward methods to overcome gradient masking. Thus, it is not clear why the authors did not try non-gradient attacks before jumping to a complicated algorithm. At the very least, those attacks (like Boundary++) should at least be part of benchmarks for comparison.
  - For starters, please refer to 'Reliable evaluation of adversarial robustness with an ensemble of diverse parameter-free attacks': they have a [publicly available implementation](https://github.com/fra31/auto-attack ) as well
  - All of this is crucial, especially since the paper claims (Section 4) that "we would like to point out that the focus of this paper is to improve gradient-based attacks on already trained BNNs"

- In general, investigating if a model exhibits gradient masking is not a contribution: standard checks like comparing transfer rates, multi-step to single-step performance, attack rates for increasing attack budgets, etc are often used to check for gradient masking.

- Figure 1: For which norm are these numbers reported? Without knowing the norm, it is hard to say if Figure b is a sign of gradient masking or not.

- Section 2.2 "..these attacks have been further strengthened by a random initial step". This is partially true: the real benefit comes from having multiple random restarts. Having just one random initialization by itself is not that useful. Please re-run evaluation experiments with random restarts (20 is a good number).

- Section 3: What does "adversarial accuracy" refer to? Is it accuracy on perturbed inputs f(x') = y, or success rate of the adversary when trying to change predictions aka f(x) ~= f(x')? Please clarify

- Section 3.1 "... clearly indicate gradient masking issues.." please elaborate: not every reader will be familiar with the set of checks used for gradient masking.

- Issues with Cross-Entropy based loss and how they promote certain magnitudes of logit values are not new. The authors might want to have a look at Section 4.1 of [Reliable Evaluation of Adversarial Robustness with an Ensemble of Diverse
Parameter-free Attacks](https://arxiv.org/pdf/2003.01690.pdf) to see if there are similarities/differences in the proposed temperature-based variant, and how the proposed method is better than the one in the Difference of Logits-Ratio based loss? This work seems to be a key and relevant part of related work and should be included in comparisons/benchmarking.

- "implementation of our algorithm will be released upon publication". Please anonymize and attach the code in response.

- The benefit of using the proposed FGSM++/PGD++ attacks on full-precision models trained with adversarial robustness seems to be negligible (Table 4), and should not be overstated in results. Also, since these attacks all have random seeds, please perform experiments multiple times for statistical significance and report summary statistics.

##########################################################################

Minor Edits:

- Section 2.2 "..perturbations to the images..." the definition here is for adversarial examples in general, and should thus be "perturbations to data"

- Section 2.2 "Gradient-based attacks can be... written as Projected Gradient Descent (PGD)" this is true only for first-order gradient-based attacks, not all gradient-based attacks (examples JSMA). Please correct.

- Section 4.1 "...since most of the modern networks consist of ReLU nonlinearities" this can (and often is) circumvented using Fake-ReLU. Example implementation [here](https://github.com/MadryLab/robustness/blob/89bdf8088a8f4bd4a8b86925a2801069ec281fee/robustness/tools/custom_modules.py#L5)

- Section 5 "...and they hypothesize that linear networks would be robust to adversarial attacks." this is not their conclusion, and seems to be out of context.

- Section 6 should preferably be either towards the end or at the beginning? Not clear why it is in the middle of other sections


Please address and clarify the cons above

---

> ### Author Response · Authors · 2020-11-18
> **AR4: Thank you for the constructive and useful feedback.  {Response to AR4 [1/2]}**
>
> We really appreciate that the reviewer finds the improvement in attack success rates by our proposed modifications especially on single step attacks like FGSM as an exciting result and also Jacobian and Hessian based proposed temperature scaling analysis to be quite insightful and interesting. Below we address the reviewer’s concerns. If any answer needs further clarification to the reviewer, we'll make the best effort to improve the clarity.
>
> ### Comparisons against black box, adaptive PGD and gradient free attacks
> - In our robustness evaluation of BNNs, we evaluate black box transfer attacks (Please refer to Figure 1 (c) and Section 3.1) on BNNs and although they perform better than original form PGD attacks under $L_\infty$ bound in some cases, they are unable to achieve 0% adversarial accuracy whereas our proposed variants achieve 0% adversarial accuracy.
> - As requested by the reviewer, we compare our method against APGD with DLR loss [a] and gradient free Square Attack [b] on different networks trained using ResNet18 and VGG16 on CIFAR10 dataset and the results are as follows:
> For ResNet-18:
> |Methods|APGD(DLR) [a]|Square Attack [b]|PGD++(NJS)|PGD++(HNS)|
> |-|-|-|-|-|
> |REF|**0.0**|0.55|**0.00**|**0.00**|
> |REF(adv. train)|49.00|54.05|**47.17**|48.54|
> |BNN-WQ|**0.00**|0.41|**0.00**|**0.00**|
> |BNN-WAQ|6.32|21.45|**0.03**|0.04|
>
>  For VGG-16:
> |Methods|APGD(DLR) [a]|Square Attack [b]|PGD++(NJS)|PGD++(HNS)|
> |-|-|-|-|-|
> |REF|0.79|2.25|**0.00**|**0.00**|
> |BNN-WQ|8.23|1.98|**0.00**|**0.00**|
> |BNN-WAQ|0.38|16.67|**0.01**|0.02|
>
> - It can be clearly seen that our proposed variants perform much better than both APGD(DLR loss) and Square Attack, consistently achieving 0% adversarial accuracy. Infact much costly gradient free Square attack is unable to achieve 0% adversarial accuracy in any of the cases under the enforced $L_\infty$ bound. The attack parameters for this experiment are the same as reported in the paper. For above experiments, we use publicly available implementation of [a].
>
> ### Identifying gradient masking in BNNs as contribution
> - We agree with the reviewer that the community has adopted these checks to identify gradient masking issues in different defense mechanisms. However, the point of our evaluation is to _eradicate the confusion in the existing literature_ [c,d,e] whether BNNs are actually robust or possess gradient masking issues since it is clearly absent in literature discussing BNNs robustness. Our main contribution after illustrating gradient masking issues in Section 2 is to improve the existing gradient based attacks using proposed two variants of finding effective scalar $\beta$ for temperature scaled attacks by analyzing the signal propagation of trained BNNs.
>
> ### BNNs Robustness Evaluation experiments with 20 random restarts
> - We agree with the reviewer that it has been previously shown that multiple random restarts makes the PGD attack much stronger. Though in this paper we have used a single random initialization step for all the experiments on FGSM/PGD/FGSM++/PGD++ to keep the comparisons fair. Apart from that, it is clearly shown in Table 2 and Table 3, that our proposed variants of PGD++ with just a single random initialization step can achieve near perfect attack success rate on almost all BNN-WQ and BNN-WAQ.
> - Here for sake of completion, we report PGD adversarial accuracies with 20 random restarts as reported in Table 1 on BNNs trained on CIFAR-10.
> |Network|ResNet-18(Rand. Restart=1)|ResNet-18(Rand. Restart=20)|VGG-16(Rand. Restart=1)|VGG-16(Rand. Restart=20)|
> |-|-|-|-|-|
> |BNN-WQ|26.98|17.91|47.32|38.49|
> |BNN-WAQ|8.57|1.94|78.01|59.26|
> - As it can be seen, with 20 random restarts although PGD adversarial accuracy does decrease but still does not reach zero, whereas using our proposed variants of PGD++, we can achieve 0% adversarial accuracy with a single random restart. It is not surprising to achieve better adversarial accuracies with several random restarts since with several random initializations, it is a high probability to encounter an initialization point with incorrect decision or low confidence on the correct class as such a point does exists.
>
> [a] Francesco Croce, Matthias Hein. "Reliable evaluation of adversarial robustness with an ensemble of diverse parameter-free attacks". In ICML 2020.
>
> [b] Maksym Andriushchenko, Francesco Croce, Nicolas Flammarion, Matthias Hein. Square Attack: a query-efficient black-box adversarial attack via random search. In ECCV 2020.
>
> [c] Angus Galloway, Graham W. Taylor, and Medhat Moussa. Attacking binarized neural networks. In ICLR, 2018.
>
> [d] Defensive quantization: When efficiency meets robustness. In International Conference on Learning Representations, 2019.
>
> [e] Sanchari Sen, Balaraman Ravindran, and Anand Raghunathan. Empir: Ensembles of mixed precision deep networks for increased robustness against adversarial attacks. In ICLR, 2020.

---

> > ### Author Response · Authors · 2020-11-18
> > **Other clarifications and suggested minor edits. {Response to AR4 [2/2]}**
> >
> > ### Issues with Cross Entropy loss and Comparison with DLR loss
> > - Thanks for pointing out this work. We would like to clarify here that cross-entropy loss is only a partial reason for poor signal propagation issue on already trained models. As explained in the paper in Section 5, for gradient based attacks to work well, both conditions should be satisfied 1) Input-Output Jacobian should be well-conditioned, 2) Error signal should be non-zero. An alternative to cross-entropy loss like Difference of Logits-Ratio (DLR) [a] based loss only tries to tackle the issue of saturating error signals which is clearly shown in our experiments using DLR loss.
> > - We show comparisons of our proposed FGSM++/PGD++ against FGSM/PGD attack under $L_\infty$ bound performed with DLR loss [a] instead of cross-entropy loss on ResNet-18 models trained on CIFAR-10 dataset. The attack parameters are the same as mentioned in the paper for all the attacks mentioned here.
> > |Methods|FGSM(DLR)|FGSM++(NJS)|FGSM++(HNS)|PGD(DLR)|PGD++(NJS)|PGD++(HNS)|
> > |-|-|-|-|-|-|-|
> > |REF|19.48|5.55|**5.35**|**0.00**|**0.00**|**0.00**|
> > |BNN-WQ|19.72|3.46|**2.51**|**0.00**|**0.00**|**0.00**|
> > |BNN-WAQ|41.78|19.46|**19.09**|4.57|**0.03**|0.04|
> > |REF(adv. train)|66.39|61.43|**61.40**|49.73|**47.17**|48.54|
> > |BNN-WQ(adv. train)|59.14|54.74|**53.82**|41.42|**40.76**|40.67|
> > - It can be clearly observed that in almost all cases *our proposed variants are much better than FGSM/PGD performed with DLR loss*. The margin of difference is significant in case of FGSM attacks and adversarial trained models. Infact, it is important to note that gradient based attacks with DLR loss perform worse on adversarially trained models than the original form of gradient based attacks.
> >
> > ### Average Statistics of PGD++ on Adversarially trained models
> > - The difference between adversarial accuracy of FGSM and FGSM++ is >1 % which is not negligible considering the fact that these are adversarially trained models and any improvement is hard to achieve in the literature. We agree that this improvement is not significant but it is definitely interesting and worth noting.
> > - As per the reviewer’s suggestion, we evaluate the adversarially trained models (trained on ResNet18 using CIFAR10) for 5 runs and report mean and standard deviation (in brackets) of adversarial accuracies here.
> > |Methods|FGSM|FGSM++(NJS)|FGSM++(HNS)|PGD|PGD++(NJS)|PGD++(HNS)|
> > |-|-|-|-|-|-|-|
> > |REF|62.41(0.031)|61.49(0.0591)|**61.46 (0.0404)**|48.79(0.0292)|**47.24(0.0440)**|48.52(0.0440)|
> > |BC|53.93(0.0521)|52.81(0.0708)|**52.23(0.0483)**|41.33(0.0203)|**39.34(0.020)**|39.92 (0.0519)|
> > - It can be clearly observed that the variance in the results over different runs is negligible and our FGSM++/PGD++ mean adversarial accuracies are better(lower) than original FGSM/PGD attacks consistently by a margin of around 1% (which is definitely interesting).
> >
> > ### Minor clarifications
> > - As mentioned in the start of Section 2, our robustness evaluation experiment of BNNs is done using $L_\infty$ bound and other attack parameters are also reported. Nevertheless, we will clarify this in the paper in both Table 1 caption and Figure 1 caption.
> > - The adversarial accuracy is the accuracy on the perturbed data. Thus, for an attack to perform better, lower adversarial accuracy is better. We will clarify this in the paper in the updated version.
> > - We will release our PyTorch implementation at the earliest once cleaned to make it understandable for users in the open-source community. Nevertheless, we provide algorithm/pseudocode for both NJS and HNS variants in the Appendix.
> >
> > ### Suggested Minor Edits
> > Thank you for these suggestions. We will make the necessary changes as mentioned below in the updated version.
> > - We will change  "..perturbations to the images..." to "..perturbations to the data..." and "Gradient-based attacks can be... written as Projected Gradient Descent (PGD)" to "First order gradient-based attacks can be... written as Projected Gradient Descent (PGD)" in the updated version of our paper.
> > - We will move the “Related works” section to the end of the paper.
> > - We are not sure the point reviewer is trying to convey when mentioning “Fake-ReLU”.
> >
> > [a] Francesco Croce, Matthias Hein. "Reliable evaluation of adversarial robustness with an ensemble of diverse parameter-free attacks". In ICML 2020.

---

> > > ### Comment · AnonReviewer4 · 2020-11-19
> > > **Most issues addressed**
> > >
> > > Thanks for the authors for a detailed response, and running suggested experiments. Observing the performance of the proposed approach under these conditions certainly is reassuring of the fact the proposed attack is better than alternatives to gradient-based attacks (along with modified versions like PGD++). These results will definitely help me better evaluate the utility of this submission :)
> > >
> > > With regards to random restarts: I would urge the authors to include numbers for 20 random restarts (like the ones you have in the replies above) instead of 1, since that restart-based randomness is an essential part of the PGD attack. Additionally, I think the decrease in performance with more restarts is much more pronounced than it is for other models. Usually, drops vary in 5-6% range, but in this case the drops are quite steep: eg 78 to 59, 8.5 to 1.9. I am curious to see how further it drops of 50 restarts are used. The authors may run it for fewer iterations, if constrained by resources: from what I can tell, the attack will benefit from more restarts than it will from more iterations.
> > >
> > > Additionally, please make sure the above experimental results (non-gradient attacks, etc) are included in your main draft: in the appendix, if the main paper does not have enough space.
> > >
> > > Also, please attach the code (in whatever form it is). The purpose for this is not to make it available to the open source community, but rather check the implementation to make sure there are no unintentional bugs. It is always better to check it at this stage, rather than realizing there was a mistake after the paper has been accepted (even though this is highly unlikely).
> > >
> > > P.S. By the "fake-ReLU" comment, what I meant was that the problem of "ReLU induced non-linearities" during back-propagation while finding adversarial examples can often be circumvented by using a 'fake-ReLU' activation instead: pretending the ReLU activation is just an identity mapping in the backward pass.

---

> > > > ### Author Response · Authors · 2020-11-23
> > > > **AR4: Thank you for acknowledging that most of the issues have been resolved.**
> > > >
> > > > We really appreciate that the reviewer finds our response was able to resolve the issues and concerns the reviewer had and the reviewer is reassured of the fact that our proposed variants are better than alternative gradient based/gradient free attacks. We would like to thank the reviewer again for suggesting the said comparisons which have surely made our paper even stronger.
> > > >
> > > > ### Submission updated according to the suggestions by the reviewer:
> > > > - We have added results with random restarts (20) in Table 1 of the paper evaluating BNNs robustness. We also put a comment in Section 6.1 (end of 2nd paragraph) specifically mentioning random restarts can improve FGSM/PGD attack but to keep comparisons fair we use single random restart.
> > > >
> > > > - We have added experimental comparisons with FGSM and PGD attack using DLR loss [a] in Table 5.
> > > >
> > > > - We have added experimental comparisons against recently introduced adaptive APGD with DLR loss [a] and gradient free Square attack [b] in Table 7.
> > > >
> > > > - We have also done all the minor edits as requested by the reviewer.
> > > >
> > > > ### Code sharing
> > > > - We are unable to share the code at the moment since cleaning the code is part of the approval process of code sharing in our organization. Since code submission was not a requirement for ICLR’21, we did not plan to initiate this approval process and we won’t be able to complete this within the discussion phase.
> > > >
> > > > - We really appreciate that the reviewer specifically mentioned the reason to share the code. We assure the reviewer that our implementation has been thoroughly tested before the submission itself to avoid any unlikely embarrassment for ourselves and unproductive time for the reviewers. In fact, we have tested and reproduced almost all the baselines using the popular Foolbox [c] library. Implementation of our method has also been extensively tested and used by other colleagues in some of their projects.
> > > >
> > > > - As promised in the manuscript, we will make sure to release the code upon the publication of the paper to ensure reproducibility.
> > > >
> > > >
> > > > ### PGD with 50 random restarts
> > > > - As requested by the reviewer, we ran some more experiments to see how well the PGD attack can do on BNNs with 50 random restarts. As suggested by reviewer, we reduced attack iterations to 10 due to resource constraints and the results are as below:
> > > > |Method|Resnet-18|VGG-16|
> > > > |-|-|-|
> > > > |BNN-WQ|16.90|39.63|
> > > > |BNN-WAQ|2.08|55.45|
> > > >
> > > > - It can be seen that 50 random restarts does not help the PGD attack drastically compared to 20 random restarts. In some cases adversarial accuracy increased whereas in some it decreased.
> > > >
> > > > - It is our view that increasing the number of random restarts is more like a heuristic random search, while our modifications to gradient based attacks is grounded on the understanding of signal propagation issues in BNNs and alleviating them in a principled way. As expected, our variants are effective even with a single random restart and obtain near perfect success rate in attacking BNNs.
> > > >
> > > > ### Fake ReLU
> > > > - Thanks for clarifying your comment. Our statement regarding ReLU was used in a different context and is also true for Fake-ReLU and we also mentioned in the brackets that our statement is true for all positive homogeneous functions.
> > > >
> > > > ### References
> > > > [a] Francesco Croce, Matthias Hein. "Reliable evaluation of adversarial robustness with an ensemble of diverse parameter-free attacks". In ICML 2020.
> > > >
> > > > [b] Maksym Andriushchenko, Francesco Croce, Nicolas Flammarion, Matthias Hein. Square Attack: a query-efficient black-box adversarial attack via random search. In ECCV 2020.
> > > >
> > > > [c] Jonas Rauber, Wieland Brendel, and Matthias Bethge. Foolbox: A python toolbox to benchmark the robustness of machine learning models. arXiv, 2017.

---

### Official Review · AnonReviewer3 · 2020-10-28
**This paper brings the temperature scaling method to attack generation mainly on BNN.**

**Rating:** 5
**Confidence:** 4

**Review:**

The paper identifies the gradient vanishing issue in the robustness of binary quantized networks. Therefore, it proposes to use temperature scaling approach in the attack generation. It has two methods for the temperature scale: (1) singular values of the input-output Jacobian and (2) maximizing the norm of the Hessian of the loss.

------------Updates after rebuttal-------------
Thanks the authors for answering my questions. However, I don't think my comments are well addressed. Even though the paper [d] may not provide public available code, the authors could either use results from the paper [d] or implement the proposed attack on the models used by [d] to see the difference.

Strengths:

+ The proposed method work well on adv trained models and floating-point models.

+ Practical approach by a simple modification to existing gradient based attacks.


Weaknesses:

- Binary quantization is not a well accepted method, since it can in general introduce >5% accuracy loss. There are a lot more valuable quantization schemes to investigate such as low-bit-with fixed point, power of 2, and additive power of 2 (Y. Li, X. Dong, and W. Wang, “Additive powers-of-two quantization: An efficient non-uniform discretization for neural networks,” in International Conference on Learning Representations, 2020).

- The novelty is limited, since it brings the temperature scaling approach, an existing method, to the problem of attacking binary quantized models.

- The paper writing is not constructed for easy understanding.


Comments and questions:

1. I would like to see comparisons with other attacks that are particularly designed for quantized models.

2. The third paragraph in Introduction, the paper tries to justify two techniques, but they are still not well motivated.

3. The fourth paraph in Introduction mentions both full precision networks and floating-point networks. What’s the difference between these two?

4. Table 1 results are surprising. Is the same observation made by other ref works?

5. The method is to replace the softmax with a monotonic function (softmax with a single scalar) during the attack generation. Then for testing the attack success rate, I think the neural network should still use the original softmax (without scalar). Then the attack success rate won’t be degraded?

---

> ### Author Response · Authors · 2020-11-18
> **AR3: Thank you for the constructive feedback.  {Response to AR3 [1/2]}**
>
> We appreciate that the reviewer finds that our simple modification to existing gradient based attacks using temperature scaling by improving the signal propagation to be effective and practical. Below we address the reviewer’s concerns. If any answer needs further clarification to the reviewer, we'll make the best effort to improve the clarity.
>
> ### Binary quantization is not a well accepted method
> - The other reviewers agree that robustness of BNNs is not well studied in the literature and is an important research direction (see AR2). We would like to add that binary network quantization is an active research area and the results have been improved to a large extent [a]. For CIFAR-10/100, BNNs achieve close to full precision network performance. Furthermore, there exist some studies on analysing BNNs robustness [b,c] but as discussed in the related works, they lack strong consensus and solid reasoning on failure of gradient based attacks. Therefore, we believe understanding the robustness properties of BNNs is important and our work makes a significant step by revealing fake robustness of BNNs and introducing two variants to improve the existing gradient based attacks by improving signal propagation. Even though we tested on the extreme case of binary quantization, we believe that our proposed variants will be equally effective for multi-bit quantization as well.
>
> ### Novelty and Paper Writing
> - As acknowledged by the other reviewers, our method constitutes an important factor the community hasn't taken notice of (AR5), *specifically the issue of poor signal propagation affecting gradient based attacks in BNNs*. We agree temperature scaling is an existing method but to the best of our knowledge _there are no existing works on using temperature scaling effectively to improve gradient based attacks_. We have also shown in Table 4 in the paper that a heuristic way of using temperature scaling for adversarial attacks performs poorly whereas our proposed two variants of choosing effective scalar $\beta$ for improving the existing gradient based attacks achieves near perfect success rate on BNNs and also show improvements on full precision and adversarially trained networks.
> - It has been a common consensus amongst other reviewers that our approach is easy to follow, and well explained. Nevertheless, we welcome any suggestions from the reviewer in improving any particular sections of the paper to make it more reader-friendly.
>
> ### Other Questions
> > I would like to see comparisons with other attacks that are particularly designed for quantized models.
>
> To the best of our knowledge, except [d], there are no specifically designed attacks for BNNs and we were unable to find a publicly available implementation of [d]. Nevertheless, [d] proposed a complex combinatorial attack for activation quantized networks but does not identify the actual issue why existing most widely used gradient based attacks like FGSM, PGD fail on BNNs. We answered this question in the paper and also proposed two simple and effective modifications which can be applied easily with minimal computational overhead and obtain near perfect success rate. The effectiveness of our variants suggests that there is little need to design an attack specifically for BNNs.
>
> >The third paragraph in Introduction, the paper tries to justify two techniques, but they are still not well motivated.
>
> Our two variants are motivated based on improving the signal propagation of the trained BNNs without changing the decision boundary. The first approach namely NJS works on conditioning the Jacobian of the network as well as improving the error signal of the network and the second one, namely HNS is based on increasing the non-linearity of the network by maximizing the Hessian norm. We have mentioned this in the fourth paragraph of introduction in the updated version of the paper.
>
> >The fourth paraph in Introduction mentions both full precision networks and floating-point networks.  What’s the difference between these two?
>
> Thanks for the suggestion. We have changed this in the updated version of our paper. Both floating point networks and full precision networks refer to traditional neural networks with both parameters and activations in real value i.e. float32.
>
>
> [a] Zechun Liu, Zhiqiang Shen, Marios Savvides, and Kwang-Ting Cheng. Reactnet: Towards precise binary neural network with generalized activation functions. In ECCV, 2020.
>
> [b] Angus Galloway, Graham W. Taylor, and Medhat Moussa. Attacking binarized neural networks. In ICLR, 2018.
>
> [c]  Defensive quantization: When efficiency meets robustness. In International Conference on Learning Representations, 2019.
>
> [d] Elias B Khalil, Amrita Gupta, and Bistra Dilkina. Combinatorial attacks on binarized neural networks. In ICLR, 2019.

---

> > ### Author Response · Authors · 2020-11-18
> > **Minor Clarifications  {Response to AR3 [2/2]}**
> >
> > ### Other Questions
> > > Table 1 results are surprising. Is the same observation made by other ref works?
> >
> > Yes, a similar observation was also hinted in [b]. As explained in this paper, this occurs due to poor signal propagation in trained BNNs and just accounts for fake robustness.
> >
> > > The method is to replace the softmax with a monotonic function (softmax with a single scalar) during the attack generation. Then for testing the attack success rate, I think the neural network should still use the original softmax (without scalar). Then the attack success rate won’t be degraded?
> >
> > Temperature scaling is only used during generation of adversarial samples. For evaluation of the adversarial samples, we use the original form of softmax without any scalar. Since the adversarial accuracies with our proposed attack variants is lower than the original form of PGD attacks, it is evident that our modified gradient based attacks are more effective in generating adversarial samples.
> >
> > [b] Angus Galloway, Graham W. Taylor, and Medhat Moussa. Attacking binarized neural networks. In ICLR, 2018.

---

### Official Review · AnonReviewer1 · 2020-10-29
**Official Blind Review**

**Rating:** 5
**Confidence:** 3

**Review:**

This paper studies the robustness of quantized neural networks against adversarial attacks. The authors use some slight modification of existing methods to successfully increase the attack success rate. In general, I think the idea is interesting. But I have some concerns that need to be addressed:
1. I am not fully convinced by the arguments made at the beginning of Section 4. The authors claim that poor signal propagation such as gradient vanishing or gradient exploding should be a problem for adversarial attacks. However, I do not think the reasonings provided here is specifically for binarized neural networks. Equations (3) and (4) also works in regular full precision networks. I do not think there are any problems which only present in BNNs, so the arguments here are not strong enough. If poor signal propagation is a problem for attacks, why we don't see that in full precision networks? More discussions on this are welcomed.
2. ResNet and DenseNet REF models in Table 3 seem to be surprisingly robust under PGD L2 attacks (column 3). This adversarial accuracy seems to be comparable to models with adversarial training. I think the authors need to provide some explanation on this.
3. (Minor) Please refrain from only using color to distinguish curves/bars in figures as it may not be friendly to readers with color blindness.
4. (Minor) The authors may need to re-organize some sections to make the paper easier to follow, for example, the "related works" section before the "experiments" section.

---

> ### Author Response · Authors · 2020-11-18
> **AR1: Thank you for the constructive feedback.**
>
> We appreciate that the reviewer finds our improved gradient based attacks using temperature scaling by improving the signal propagation based on network Jacobian and Hessian norm to be interesting and successful in increasing attack success rate. Below we address the reviewer’s concerns. If any answer needs further clarification to the reviewer, we'll make the best effort to improve the clarity.
>
> ### Poor Signal Propagation Issues in BNNs
> - We would like to clarify that the main difference between binary networks and floating point networks is regarding the parameter and activation values. The weight distribution in trained BNNs is very different from the initialization due to the binary constraint on the weights. This heavily affects the signal propagation properties of BNNs as observed in terms of large mean Jacobian singular values in Table 12 in the Appendix. This consequently causes gradient vanishing issues in trained BNNs. Whereas floating point networks are initialized to have good signal propagation (refer [a]) which is maintained approximately even at the end of training (refer Table 12). Thus, full precision networks do not exhibit fake robustness and are easily attacked using the original gradient based attacks.
>
> ### Clarification on adversarial accuracies obtained using PGD $L_2$ attacks
> - Adversarial samples under $L_2$ bound are much harder to estimate compared to $L_\infty$ bound. It is common in literature to have non-zero adversarial accuracy with PGD attack under $L_2$ bound (refer to Table 1 in [b]). We have evaluated adversarial accuracies with attack radius 120/255 similar to [b]. If the $L_2$ bound is increased further, the adversarial accuracies of full precision models can potentially reach to zero. Also, since PGD attack under $L_2$ bound uses the input gradients itself instead of the sign of input gradients, change in step size and number of iterations can affect the adversarial accuracy for full precision models. This does not imply full precision models are robust under $L_2$ bound and neither do we claim this anywhere in the paper.
> - We have used standard hyperparameters for $L_2$ PGD attack as mentioned in FoolBox library [c] and our attack parameters have been provided in Table 6 in the Appendix. The point of this comparison of our modified attacks against PGD attack under $L_2$ bound is to show consistent improvement of our method under $L_2$ bound on adversarial samples even when using the same set of hyperparameters for both PGD and our modified PGD++.
>
> ### Minor Edits
> Thank you for these suggestions. We have changed the curve/bars in the updated version of the paper to make it more reader-friendly. We have also moved the “related works” section after "Experiments" section in the updated version.
>
> [a] Xavier Glorot and Yoshua Bengio. Understanding the difficulty of training deep feedforward neural networks. In AISTATS 2010.
>
> [b] Chris Finlay, Aram-Alexandre Pooladian, and Adam Oberman. The log barrier adversarial attack: making effective use of decision boundary information. In ICCV, 2019.
>
> [c] Jonas Rauber, Wieland Brendel, and Matthias Bethge. Foolbox: A python toolbox to benchmark the robustness of machine learning models. arXiv preprint arXiv:1707.04131, 2017.

---

### Official Review · AnonReviewer2 · 2020-11-05
**The paper studies the robustness of binary neural networks. It highlights the issue of signal propagation in BNNs. To mitigate this issue, the authors propose a temperature rescaling technique.**

**Rating:** 6
**Confidence:** 5

**Review:**

**Update**: Thanks to the authors for addressing my comments. As it was pointed out by the authors, temperature rescaling is mostly applicable to non-linear loss functions. For linear loss functions, temperature scaling only linear rescales the gradients. The difference between the proposed PGD++ attack and PGD with linear DLR loss is small (see the author's response to AR4). The improvements are most significant for FGSM but FGSM is not recommended for the robustness evaluation. Given the limited technical novelty and small improvements for linear loss functions, my score remains unchanged.

###### Summary
The paper studies the robustness of binary neural networks (BNNS), which at first look have higher robustness than full-precious neural networks. The authors highlight the problem of poor signal propagation in BNNs, which makes gradient-based attacks difficult. To address this issue, the authors proposed a 1) single scalar rescaling of the jacobian to improve the signal propagation; 2) parameter-free hessian norm scaling technique. In the experiments, the authors demonstrated that the modified attacks reduce the accuracy of BNNs to zero and outperform existing gradient-based attacks against floating-point networks.

###### Reasons for score

I vote for a weak acceptance of this paper. The paper shows that BNNs are not robust and introduce an interesting gradient rescaling technique, which can also be used to attack full-precision networks. The rescaling technique is well explained, easy to apply for any existing attacks, and has low computational overhead. However, as I will discuss below, I see some problems comparing the proposed attack against a well-tuned PGD attack.

###### Pros:
1) The paper studies the robustness of BNNs. The robustness of BNNs is not well studied, and understanding the robustness of BNNs is an important research direction.
2) The authors highlight the issue of signal propagation in BNNs. To address this issue, they devise a novel low-computational complexity technique.
3) Experimental results for BNNs and full-precious models demonstrate that the modified attack is effective.

###### Concerns and questions:
- In the experiments, the authors use a single parameter for the step size for both PGD and FGSM attacks. On the other hand, the proposed method computes an optimal rescaling to achieve good signal propagation. Even though the proposed technique has a low computational budget, I believe the authors should do a grid search for the optimal step size for PGD and FGSM attacks for a fair comparison.
- In the experiments, PGD L2 attack was unable to reduce the naturally trained models' accuracy to 0. This seems strange and unlikely as gradient shattering should not happen for naturally trained models. Can the authors explain these results? Is it possible that there might be an implementation issue?

###### Comments and suggestions:
- The proposed technique amplifies the error signal in a nonlinear way for nonlinear losses such as cross-entropy. However, for other losses such as multiclass-hinge loss, CW loss, the proposed method will simply linearly rescale the error signal. Attacking CW loss might be useful as it avoids the issues of saturated softmax gradients. Will this technique be useful for attacking with CW loss?
- The authors claim that this method improves the efficiency of white-box attacks against full precision models. Is it possible for the authors to get the results for mnist_challenge and cifar10_challenge to see if the method outperforms an optimally tuned PGD attack?

---

> ### Author Response · Authors · 2020-11-18
> **AR2: Thank you for the positive feedback.  {Response to AR2 [1/2]}**
>
> We appreciate that the reviewer finds that our rescaling technique to improve the signal propagation is novel, interesting, well explained, and easy to apply with low computational overhead. Below we address the reviewer’s concerns. If any answer needs further clarification to the reviewer, we'll make the best effort to improve the clarity.
>
> ### Tuning the Step Size of PGD and FGSM
> - We have used the standard hyperparameters for FGSM and PGD attack as used in the literature which we have also mentioned in Table 6 and used the same hyperparameters for modified attacks to make fair comparisons. We would like to clarify here that step size $\eta$ and $\beta$ have different effects in the attacks performed. As explained in Section 2.2, PGD and FGSM attack under $L_\infty$ bound only use the sign of input gradient in each gradient ascent step. Thus, if the input gradients are completely saturated (which is the case for BNNs), original forms of PGD or FGSM will not work irrespective of the step size used. Also, we would like to mention here that since FGSM is a single step attack, step size is not a free parameter and it is the maximum allowed perturbation magnitude.
> - We performed extensive step size tuning for original form of PGD attack on different ResNet18 models trained on CIFAR10 and the adversarial accuracies are reported as follows:
> |Step Size|0.5|1.0|2.0|4.0|6.0|8.0|10.0|12.0|14.0|18.0|22.0|25.0|30.0|40.0|
> |-|-|-|-|-|-|-|-|-|-|-|-|-|-|-|
> |REF|0.0|0.0|0.0|0.0|0.01|0.03|0.06|0.45|2.41|7.42|7.42|7.42|7.42|7.42|
> |BNN-WQ|28.07|27.15|26.96|26.52|25.74|25.35|24.87|24.87|24.97|25.93|25.93|25.93|25.93|25.93|
> |BNN-WAQ|14.51|10.90|8.70|7.29|6.60|6.62|8.30|9.96|12.78|16.40|16.40|16.40|16.40|16.40|
>
> - It can be observed clearly that although tuning the step size lowers adversarial accuracy a bit in some cases but still _cannot reach zero_ for BNNs unlike our proposed variants. We have also reported these step size tuning results in Figure 4 in the updated version of our paper.
>
> ### Clarification on adversarial accuracies obtained using PGD $L_2$ attacks
> - Adversarial samples under $L_2$ bound are much harder to estimate compared to $L_\infty$ norm. It is common in literature to have non-zero adversarial accuracy with PGD attack (refer to Table 1 in [b]) under $L_2$ bound. We have evaluated adversarial accuracies with attack radius 120/255 similar to [b]. If the $L_2$ bound is increased further, the adversarial accuracies of full precision models can potentially reach to zero. Also, since PGD attack under $L_2$ bound uses the input gradients itself instead of the sign of input gradients, change in step size and number of iterations can affect the adversarial accuracy for floating point models. This does not imply full precision models are robust under $L_2$ bound and neither do we claim this anywhere in the paper.
> - We have used standard hyperparameters for $L_2$ PGD attack as mentioned in FoolBox library [c] and our attack parameters have been provided in Table 6 in the Appendix. The point of this comparison of our modified attacks against PGD attack under $L_2$ bound is to show consistent improvement of our method under $L_2$ bound on adversarial samples even when using the same set of hyperparameters for both PGD and our modified PGD++.
>
> ### Our techniques on other loss functions
> - As illustrated in the paper, our proposed variants NJS and HNS do not only focus on improving the error signal but also on improving the conditioning of the Jacobian in NJS and similarly HNS focuses on increasing the non-linearity. Thus, our rescaling technique is independent of the loss function and it could potentially improve adversarial attacks even on other loss functions. We would like to point out that our approach is effective *only if there exists an issue of gradient vanishing* (ie, fake robustness) and if linear loss functions (such as Hinge loss or CW loss) do not exhibit fake robustness, the improvements with our method would be marginal.
> - We have also compared our method against FGSM/PGD with DLR loss proposed in [a] and our proposed variants outperform FGSM/PGD attack performed using DLR loss as reported in response to AR4. We have also reported these results in Table 5 of our updated manuscript.
>
> [a] Francesco Croce, Matthias Hein. "Reliable evaluation of adversarial robustness with an ensemble of diverse parameter-free attacks". In ICML 2020.
>
> [b] Chris Finlay, Aram-Alexandre Pooladian, and Adam Oberman. The log barrier adversarial attack: making effective use of decision boundary information. In ICCV, 2019.
>
> [c] Jonas Rauber, Wieland Brendel, and Matthias Bethge. Foolbox: A python toolbox to benchmark the robustness of machine learning models. arXiv preprint arXiv:1707.04131, 2017.

---

> > ### Author Response · Authors · 2020-11-18
> > **Other questions. {Response to AR2 [2/2]}**
> >
> > ### Evaluation on CIFAR10 challenge
> > - Our implementation is written in PyTorch and cifar10/mnist challenge only has models and architecture written in Tensorflow. We will evaluate our proposed variants on this challenge once we port our code to Tensorflow. Nevertheless, we used standard attack parameters as used in literature and thus we believe our experimental observations should replicate on the challenge as well.

---

### Official Review · AnonReviewer5 · 2020-11-06
**Interesting observation from quantized networks results in general, stronger adversarial attacks**

**Rating:** 7
**Confidence:** 4

**Review:**

This work starts by questioning the apparent robustness of quantized networks and demonstrates that such robustness is more so a failure of the attack algorithm in picking up the gradient signal. The authors address this by tuning a scalar multiplier applied to the network logits, which doesn’t modify the model’s decision boundary. Through analyzing the Jacobian, two approaches are proposed to determine the scalar $\beta$ without tuning it by performing the attack. This approach is quite effective on quantized networks and even provides significant improvement on floating-point networks combining with existing attacks like FGSM and PGD. The proposed modification might seem trivial at first, but it constitutes an important factor the community hasn’t taken notice of, to the best of my knowledge.

A few questions: 1) I don’t see any mentions of tuning the attack step size; if we set the new $\eta$ to $\eta/\beta$, we can keep the Jacobian intact and isolate the effect of temperature scaling for XENT. 2) how about sweeping $\beta$ and plotting against adversarial accuracy? This is surely expensive, but it would paint a clearer picture of the optimality of $\beta$ found using the proposed approaches. This can be done in tandem with an attack step of $\eta/\beta$.

I like the result overall, though the effect of $\beta$ on the Jacobian and the softmax can and should be separated, if my understanding is correct. The proposed approaches for determining $\beta$ largely depend on the Jacobian; therefore, there should be more investigation on if scaling the Jacobian correctly is more important than getting good error signals from softmax.

---

> ### Author Response · Authors · 2020-11-18
> **AR5: Thank you for the positive feedback.**
>
> We appreciate that the reviewer finds our approach to improve the signal propagation quite effective and novel. Below we address the reviewer’s concerns. If any answer needs further clarification to the reviewer, we'll make the best effort to improve the clarity.
>
> ### Tuning the Step Size of PGD and FGSM
>
> - We have used the standard hyperparameters for FGSM and PGD attack as used in the literature which we have also mentioned in Table 6 and used the same hyperparameters for modified attacks to make fair comparisons. We would like to clarify here that step size $\eta$ and $\beta$ have different effects in the attacks performed. As explained in Section 2.2, PGD and FGSM attack under $L_\infty$ bound only use the sign of input gradient in each gradient ascent step. Thus, if the input gradients are completely saturated (which is the case for BNNs), original forms of PGD or FGSM will not work irrespective of the step size used. Also, we would like to mention here that since FGSM is a single step attack, step size is not a free parameter and it is the maximum allowed perturbation magnitude.
> - We performed extensive step size tuning for original form of PGD attack on different ResNet18 models trained on CIFAR-10 and the adversarial accuracies are reported as follows:
> |Step Size|0.5|1.0|2.0|4.0|6.0|8.0|10.0|12.0|14.0|18.0|22.0|25.0|30.0|40.0|
> |-|-|-|-|-|-|-|-|-|-|-|-|-|-|-|
> |REF|0.0|0.0|0.0|0.0|0.01|0.03|0.06|0.45|2.41|7.42|7.42|7.42|7.42|7.42|
> |BNN-WQ|28.07|27.15|26.96|26.52|25.74|25.35|24.87|24.87|24.97|25.93|25.93|25.93|25.93|25.93|
> |BNN-WAQ|14.51|10.90|8.70|7.29|6.60|6.62|8.30|9.96|12.78|16.40|16.40|16.40|16.40|16.40|
> - It can be observed clearly that although tuning the step size lowers adversarial accuracy a bit in some cases but still _cannot reach zero_ for BNNs unlike our proposed variants. We have also reported these step size tuning results in Figure 4 in the updated version of our paper.
>
>
> ### Sweeping $\beta$ against adversarial accuracy
> - For both the approaches, we obtain $\beta$ for each sample independently in order to maximize the signal propagation for each sample. Specifically, in our proposed HNS variant, for each sample we find a different optimal $\beta$ maximizing the Hessian norm. For the NJS variant, $\beta$ is estimated as the product of two values: 1) inverse of the mean of Jacobian singular values over a set of 100 test samples, and 2) a sample specific scalar obtained using Proposition 1. Therefore, plotting $\beta$ vs. adversarial accuracy is not trivial.
> - To illustrate that our proposed variants can achieve optimal value of $\beta$, we would like to refer to Figure 5 (b) in the paper. Since $L_\infty$ bounded attacks use sign of input gradients for the attack, the only way our proposed approach can achieve 0% adversarial accuracy is if our approach finds $\beta$ within the range of optimality for each sample as illustrated in the Figure.
> - To our understanding, both (conditioning of Jacobian and non-zero error signals) are important since if the error signal is saturated (zero) then irrespective of how well the Jacobian is conditioned, the attack will fail and vice-versa.

---

### Decision · Program_Chairs · 2021-01-07
**Final Decision**

**Decision:**

Reject

**Comment:**

The paper studies the robustness of binary neural networks (BNNS), showing how quantized models suffer from gradient vanishing. To solve this issue, the authors propose temperature scaling approaches that can overcome this masking, achieving near-perfect perfect success in crafting adversarial inputs for these models. The problem is interesting and important. However, the major concerns are that the technical novelty is limited raised by two Reviewers, small improvements for linear loss functions. The most related work is not compared in the experiment.